# Studying Exploration in RL: An Optimal Transport Analysis of Occupancy Measure Trajectories

**Reabetswe M. Nkhumise**                                    *rmnkhumise1@sheffield.ac.uk*
*Department of Computer Science, The University of Sheffield, UK*

**Debabrota Basu**                                    *debabrota.basu@inria.fr*
*Eq́uipe Scool, Univ. Lille, Inria, CNRS, France*

**Tony J. Prescott**                                    *t.j.prescott@sheffield.ac.uk*
*Department of Computer Science, The University of Sheffield, UK*

**Aditya Gilra**                                    *aditya.gilra@cwi.nl*
*Centrum Wiskunde & Informatica, Netherlands*
*Department of Computer Science, The University of Sheffield, UK*

**Reviewed on OpenReview:** *https://openreview.net/forum?id=pdC092Nn8N*

## Abstract

The rising successes of RL are propelled by combining smart algorithmic strategies and deep architectures to optimize the distribution of returns and visitations over the state-action space. A quantitative framework to compare the learning processes of these eclectic RL algorithms is currently absent but desired in practice. We address this gap by representing the learning process of an RL algorithm as a sequence of policies generated during training, and then studying the policy trajectory induced in the manifold of state-action occupancy measures. Using an optimal transport-based metric, we measure the length of the paths induced by the policy sequence yielded by an RL algorithm between an initial policy and a final optimal policy. Hence, we first define the *Effort of Sequential Learning (ESL)*. ESL quantifies the relative distance that an RL algorithm travels compared to the shortest path from the initial to the optimal policy. Furthermore, we connect the dynamics of policies in the occupancy measure space and regret (another metric to understand the suboptimality of an RL algorithm), by defining the *Optimal Movement Ratio* (OMR). OMR assesses the fraction of movements in the occupancy measure space that effectively reduce an analogue of regret. Finally, we derive approximation guarantees to estimate ESL and OMR with a finite number of samples and without access to an optimal policy. Through empirical analyses across various environments and algorithms, we demonstrate that ESL and OMR provide insights into the exploration processes of RL algorithms and the hardness of different tasks in discrete and continuous MDPs.

## 1 Introduction

In recent years, significant advancements in Reinforcement Learning (RL) have been achieved in developing exploration techniques that improve learning (Bellemare et al., 2016; Burda et al., 2019; Eysenbach et al., 2019) along with new learning methods (Lazaridis et al., 2020; Müller et al., 2021; Li, 2023). With growing computational resources, these techniques have led to various successful applications of RL, such as playing games up to human proficiency (Silver et al., 2017; Jaderberg et al., 2019), controlling robots (Ibarz et al., 2021; Kaufmann et al., 2023), tuning databases and computer systems (Wang et al., 2021; Basu et al., 2019), etc. However, there remains a lack of consensus over approaches that can quantitatively compare these exploratory processes across RL algorithms and tasks (Seijen et al., 2020; Amin et al., 2021; Ladosz et al.,

2022). This is attributed to some methods being algorithm-specific (Tang et al., 2017), while others provide theoretical guarantees for very specific settings (Lattimore & Szepesvári, 2020; Agarwal et al., 2022). Thus, comparing the exploratory processes of these eclectic algorithms across the multi-directional space of RL algorithm design emerges as a natural question. However, the present literature lacks a metric to compare them except regret, which is often hard to estimate (Ramos et al., 2017; 2018).

This paper aims to address this gap based on two key observations. *First*, we observe from the linear programming formulation of RL that solving the value maximisation problem is equivalent to finding an optimal occupancy measure (Syed et al., 2008; Neu & Pike-Burke, 2020; Kalagarla et al., 2021). Occupancy measure is the distribution of state-action pair visits induced by a policy (Altman, 1999; Laroche & des Combes, 2023). Under mild assumptions, a policy maps uniquely to an occupancy measure. *Second*, we observe that any RL algorithm learns by sequentially updating policies starting from an initial policy to reach an optimal policy. The search for an optimal policy is influenced by the exploration-exploitation strategy and functional approximators, both of which impact the overall performance of the agent by determining the quality of experiences from which it learns (Zhang et al., 2019; Ladosz et al., 2022). Hereby, we term collectively the learning strategy and the exploration-exploitation interplay as the *exploratory process.*

**Contributions.** *1. A Framework.* Motivated by our observations, we abstract any RL algorithm as a trajectory of occupancy measures induced by a sequence of policies between an initial and a final (optimal) policy. The occupancy measure of a policy given an environment corresponds to the data-generating distribution of state-actions. Thus, we can quantify the effort of each policy update, i.e. the effort to shift the state-action data distributions, as the transportation distance between their occupancy measures. The total effort of learning by the algorithm can be measured as the total distance covered by its occupancy measure trajectory. We provide a mathematical basis for this quantification by proving that the space of occupancy measures is a differentiable manifold for smoothly parameterized policies (Section 3). Hence, we can compute the length of the occupancy measure trajectory on this manifold using Wasserestein distance as the metric (Villani, 2009).

*2. Effort of Sequential Learning.* In contrast to RL, if we knew the optimal policy we could update our initial policy directly via supervised or imitation learning. Effort of this learning is represented by a direct, shortest (geodesic) path from initial to optimal policy on the occupancy measure manifold. To quantify the cost of the exploratory process to learn the environment, we define the *Effort of Sequential Learning* (ESL) as the ratio of the (indirect) path traversed by an RL algorithm in the occupancy measure space to the direct distance between the initial and optimal policy (Section 3.1). Lower ESL implies more efficient exploratory process.

*3. Efforts to learn that lead to Regret-analogue minimisation.* Regret is a widely used optimality measure for reward-maximizing RL algorithms (Jaksch et al., 2010). It measures the total deviation in the value functions achieved by a sequence of policies learned by an RL algorithm with respect to the optimal algorithm that always uses the optimal policy (Sinclair et al., 2023). We show that regret is related to the sum of distances between the optimal policy and each policy in the sequence learned by the RL algorithm, in the occupancy measure space. We can define an analogue of instantaneous regret (at any one step during learning rather than cumulative), in the occupancy measure space, as the geodesic distance between the occupancy measure of the policy at this step in the learning sequence, and the optimal one. We find that not all policy updates lead to a reduction in this analogue of immediate regret, and thus define another index *Optimal Movement Ratio* (OMR) that measures the fraction that do (Section 3.2).

*4. Computational and Numerical Insights.* We prove sample complexity guarantees to approximate ESL and OMR in practice as we do not have access to the occupancy measures but collection of rollouts from the corresponding policies (Section 4). We show the relation of empirical OMR and ESL to the true ones if the optimal policy is never reached by an algorithm. We conduct experiments on multiple environments, both discrete and continuous, with sparse and dense rewards, comparing state-of-the-art algorithms. We observe that by visualizing aspects of the path traversed (and by comparing ESL and OMR), we are able to compare and provide insights into their exploratory processes and the impact of task hardness on them (Section 5).

## 2 Preliminaries

**Markov Decision Processes.** Consider an agent interacting with an environment in discrete timesteps. At each timestep $t \in \mathbb{N}$, the agent observes a state $s_t$, executes an action $a_t$, and receives a scalar reward $\mathcal{R}(s_t, a_t)$. The behaviour of the agent is defined by a policy $\pi(a_t|s_t)$, which maps the observed states to actions. The environment is modelled as a Markov Decision Process (MDP) $\mathbb{M}$ with a state space $\mathcal{S}$, action space $\mathcal{A}$, transition dynamics $\mathcal{T} : \mathcal{S} \times \mathcal{A} \to \mathcal{S}$, and reward function $\mathcal{R} : \mathcal{S} \times \mathcal{A} \to \mathbb{R}$. During task execution, the agent issues actions in response to visited states, and hence a sequence of states and actions $h_t = (s_0, a_0, s_1, a_1, ..., s_{t-1}, a_{t-1}, s_t)$, here called a rollout, is observed. In infinite-horizon settings, the state value function for a given policy $\pi$ is the expected discounted cumulative reward over time $V_\pi(s) \triangleq \mathbb{E}_\pi \left[ \sum_{t=0}^\infty \gamma^t \mathcal{R}(s_t, a_t) \mid s_0 = s \right]$, where $\gamma \in [0, 1)$ is the discount rate. The goal is to learn a policy that maximises the objective $J_\mu^\pi \triangleq \mathbb{E}_{s \sim \mu}[V_\pi(s)]$, where $\mu(s)$ is the initial state distribution.

**Occupancy Measure.** The state-action occupancy measure is a distribution over the $\mathcal{S} \times \mathcal{A}$ space that represents the discounted frequency of visits to each state-action pair when executing a policy $\pi$ in the environment (Syed et al., 2008). Formally, the occupancy measure of $\pi$ is $v_\pi(s, a) \triangleq \rho \sum_{t=0}^\infty \gamma^t \mathbb{P}(s_t = s, a_t = a \mid \pi, \mu)$, where $\rho = 1 - \gamma$ is the normalizing factor.

Stationary Markovian policies allow a bijective correspondence with their state-action occupancy measures (Givchi, 2021). We express the objective $J_\mu^\pi$ in terms of the occupancy measure as

$$J_\mu^\pi = \frac{1}{\rho}\mathbb{E}_{(s,a)\sim v_\pi} \left[ \bar{\mathcal{R}}(s, a) \right], \tag{1}$$

where $\bar{\mathcal{R}}(s, a)$ is the expected immediate reward for the state-action pair $(s, a)$.

**Wasserstein Distance.** Let $\mu, \nu \in \mathcal{P}(\mathcal{X})$ be probability measures on a complete and separable metric (Polish) space $(\mathcal{X}, d_\mathcal{X})$. The p-Wasserstein distance between $\mu$ and $\nu$ is (Villani, 2009)

$$\mathcal{W}_p(\mu, \nu) \triangleq \left( \min_{\pi \in \Pi(\mu, \nu)} \int_{\mathcal{X} \times \mathcal{X}} c(x, x') \, d\pi(x, x') \right)^{1/p}, \tag{2}$$

where the cost function is given by the metric as $c(x, x') = (d_\mathcal{X}(x, x'))^p$ for some $p \geq 1$. $\Pi(\mu, \nu)$ is a set of all admissible transport plans between $\mu$ and $\nu$, i.e. probability measures on $\mathcal{X} \times \mathcal{X}$ space with marginals $\mu$ and $\nu$. Wasserstein distances induce geodesic in well-behaved spaces of probability measures. For more discussion, we refer to Appendix A.9. For this work, we consider 1-Wasserstein distance, i.e. $p = 1$, though the results are generalisable to $p > 1$.

**MDPs with Lipschitz Rewards.** Following Pirotta et al. (2015) and Kallel et al. (2024), we assume an MDP with $L_\mathcal{R}$-Lipschitz rewards (ref. Appendix A.1 for elaboration) that satisfies $|\bar{\mathcal{R}}(s, a) - \bar{\mathcal{R}}(s', a')| \leq L_\mathcal{R} d_{\mathcal{S}\mathcal{A}}((s, a), (s', a'))$ for all $s, s' \in \mathcal{S}$ and $a, a' \in \mathcal{A}$. Here, $d_{\mathcal{S}\mathcal{A}}((s, a), (s', a')) = d_\mathcal{S}((s, s')) + d_\mathcal{A}((a, a'))$ is the metric defined on the joint state-action space $\mathcal{S} \times \mathcal{A}$. This is a weaker condition than assuming a completely Lipschitz MDP. Pirotta et al. (2015) showed that for any pair of stationary policies $\pi$ and $\pi'$, the absolute difference between their corresponding objectives is

$$\left| J_\mu^\pi - J_\mu^{\pi'} \right| \leq \frac{L_\mathcal{R}}{\rho} \mathcal{W}_1(v_\pi, v_{\pi'}), \tag{3}$$

where $\mathcal{W}_1(v_\pi, v_{\pi'})$ is the 1-Wasserstein distance between the occupancy measures of the policies (ref. Appendix A.2 for details).

## 3 RL Algorithms as Trajectories of Occupancy Measures

The exploration process (i.e. the exploration-exploitation interplay and learning strategy) of an RL algorithm influences how the policy model updates its policies (Kaelbling et al., 1996; Sutton & Barto, 2018). During training, a *policy trajectory*, i.e. sequence of policies $(\pi_0, \pi_1, \ldots, \pi_N)$, is generated over policy updates due to the exploratory process. We assume these policies belong to a set of stationary Markov policies parameterized

by $\theta \in \Theta$. For policies in this set $\pi_\theta \in \mathbf{\Gamma}_\Theta$, we define the space of occupancy measures corresponding to $\mathbf{\Gamma}_\Theta$ as $\mathcal{M} = \{v_{\pi_\theta}(s,a) \mid \pi_\theta \in \mathbf{\Gamma}_\Theta, \theta \in \Theta\}$.

**Proposition 1** (Properties of $\mathcal{M}$). *If the policy $\pi$ has a smooth parameterization $\theta$, then the space of occupancy measures $\mathcal{M}$ is a differentiable manifold. (Proof in Appendix A.3)*

We can endow the manifold $\mathcal{M}$ with a 1-Wasserstein metric $\mathcal{W}_1$ to the compute the length of any path on $\mathcal{M}$ since $(\mathcal{M}, \mathcal{W}_1)$ is a geodesic space (ref. Appendix A.9 for details). The path distance between occupancy measures corresponding to policies parameterised by $\theta, \theta + d\theta \in \mathcal{M}$ is $ds = \mathcal{W}_1(v_{\pi_\theta}, v_{\pi_{\theta+d\theta}})$. Additionally, in imitation learning, the 1-Wasserstein distance between the occupancy measures of the learner and expert can be used as a minimisable loss function to learn the expert's policy (Zhang et al., 2020). Hence, the 1-Wasserstein distance reflects the effort required to achieve this imitation learning. Similarly, we propose the following quantification of the effort to update from one policy to another.

**Definition 1** (Effort of Learning). *We define the 1-Wasserstein metric between occupancy measures of two policies $\pi$ and $\pi'$, i.e. $\mathcal{W}_1(v_\pi, v_{\pi'})$, as the effort required to learn or update from one policy to the other.*

When a learning process causes an update between occupancy measures in $\mathcal{M}$, we attribute the resulting update effort to the learning process and refer to it as the effort of learning. In a learning process, first the initial policy $\pi_0$ is obtained typically by randomly sampling the model parameters, then these parameters $\theta$ undergo updates until a predefined convergence criterion is satisfied, yielding the final optimal policy $\pi_N = \pi^*$. Since each policy has a corresponding occupancy measure, this process yields a sequence of points on $\mathcal{M}$, which can be connected by geodesics between successive points, producing a curve. The length of the curve is computed by the summation of the finite geodesic distances between consecutive policies along it (Lott, 2008),

$$C \triangleq \sum_{k=0}^{N-1} \mathcal{W}_1(v_{\pi_{\theta_k}}, v_{\pi_{\theta_{k+1}}}), \tag{4}$$

where $\theta_0$ and $\theta_N$ are respectively the initial and final parameter values before and after learning.

### 3.1 Effort of Sequential Learning (ESL)

As we saw above, RL generates a trajectory in the occupancy measure manifold $\mathcal{M}$, whose length is given by Equation (4). Compared to the long trajectory of sequential policies generated by the exploratory process, the geodesic $L$ is the ideal shortest path to the optimal policy $\pi_N = \pi^*$ from $\pi_0$, whose length is $L = \mathcal{W}_1(v_{\pi_0}, v_{\pi_N})$. This path would be taken by an imitation-learning oracle algorithm that knows $\pi^*$. Both these paths are schematically depicted in Figure 1.

**Definition 2** (Effort of Sequential Learning (ESL)). *We define the effort of sequential learning incurred by a trajectory of the exploratory process of an RL algorithm, relative to the oracle that knows $\pi^*(= \pi_N)$ as,*

$$\eta \triangleq \frac{\sum_{k=0}^{N-1} \mathcal{W}_1(v_{\pi_k}, v_{\pi_{k+1}})}{\mathcal{W}_1(v_{\pi_0}, v_{\pi_N})} \tag{5}$$

*Due to the stochasticity of the exploratory process, we introduce an expectation to obtain $\bar{\eta} = \mathbb{E}_{\pi_0, \mu}[\eta]$. We refer to $\bar{\eta}$ as the effort of sequential learning (ESL).*

$\bar{\eta} \geq 1$ and a larger $\bar{\eta}$ corresponds to a less efficient exploratory process of the RL algorithm. Hence, an RL algorithm with $\bar{\eta} \approx 1$ closely mimics the oracle and has an efficient exploratory process.

### 3.2 Optimal Movement Ratio (OMR)

Regret measures the total deviation in value functions incurred by a sequence of policies learned by an RL algorithm with respect to the optimal algorithm that always uses the optimal policy (Sinclair et al., 2023). We show that regret is connected to the sum of distances from each policy (in the sequence learned by an RL algorithm) to the optimal policy in the occupancy measure space.

**Proposition 2** (Regret and Occupancy Measures). *Given an MDP with $L_\mathcal{R}$-Lipschitz rewards, we obtain $Regret \triangleq \sum_{k=1}^{N} \left(J_\mu^{\pi^*} - J_\mu^{\pi_k}\right) \leq \frac{L_\mathcal{R}}{\rho} \sum_{k=1}^{N} \mathcal{W}_1(v_{\pi_k}, v_{\pi^*})$. (Proof in Appendix A.4)*

We refer to $\mathcal{W}_1(v_{\pi_k}, v_{\pi^*})$ as the *distance-to-optimal*, and analogously use it as the expected immediate regret in the occupancy measure space. Furthermore, we refer to $\mathcal{W}_1(v_{\pi_k}, v_{\pi_{k+1}})$ as *stepwise-distance*. Interestingly, during training, the *distance-to-optimal* and *stepwise-distance* share a relationship illustrated in Figure 2. From Figure 2, we observe that if the change in *distance-to-optimal*, $\delta_k \triangleq \mathcal{W}_1(v_{\pi_k}, v_{\pi^*}) - \mathcal{W}_1(v_{\pi_{k+1}}, v_{\pi^*}) > 0$, it indicates that the agent got closer to the optimal. We define the set $K^+$ as containing indices $k$ for which $\delta_k > 0$, while $K^-$ contains the rest.

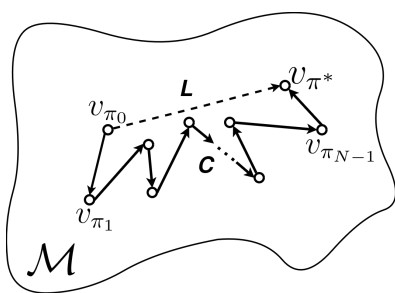

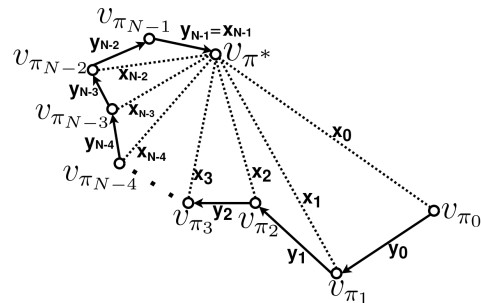

Figure 1: Schematic of the policy trajectory $C$ in the space of occupancy measures $\mathcal{M}$ during RL training (solid line) vs the geodesic $L$ (shortest path, dashed line) between the initial and final points (i.e. $\pi_0$ and $\pi_N = \pi^*$).

Figure 2: Schematic of how *distance-to-optimal* (denoted by $x_k$) and *stepwise-distance* (denoted by $y_k$) on the occupancy measure space describe exploratory process of an RL algorithm during training.

**Definition 3** (Optimal Movement Ratio (OMR)). *We define the proportion of policy transitions that effectively reduce the distance-to-optimal, in a learning trajectory, as*

$$\kappa \triangleq \frac{\sum_{k \in K^+} \mathcal{W}_1(v_{\pi_k}, v_{\pi_{k+1}})}{\sum_{k=0}^{N-1} \mathcal{W}_1(v_{\pi_k}, v_{\pi_{k+1}})} . \tag{6}$$

*Due to the stochasticity of the exploratory process, we introduce an expectation to obtain $\bar{\kappa} = \mathbb{E}_{\pi_0, \mu}[\kappa]$. We refer to $\bar{\kappa}$ as the optimal movement ratio (OMR).*

Note that $\bar{\kappa} \in [0, 1]$, and $\bar{\kappa} \to 1$ indicates that nearly all the policy updates reduce the *distance-to-optimal*, thus showing high efficiency. $\bar{\kappa} \to 0$ implies low efficiency, since only a small fraction of the policy updates contribute towards the reduction of the *distance-to-optimal*.

The definitions of ESL and OMR assume that the policy at the end of learning is optimal. In Section 4.2, we define a version of ESL that is useful for the cases where an optimal policy is not reached. While this is not an empirical proxy, we show in Section 5.3 and Appendix B.5 that it is useful when the final policy is closer to optimal than the initial one. While regret also depends on an optimal policy, it is related to cumulative rewards, whereas our metrics do not explicitly depend on rewards. Still, we show a bound with regret in Proposition 2, and further discuss the possibility of extending our metrics to be reward-aware in Section 6. We show empirically that our metrics are complementary to regret in Section 5.2, and discuss other connections with regret in Section 7.

### 3.3 Extension to Finite-Horizon Episodic Setting

In the episodic finite-horizon MDP formulation of RL, in short *Episodic RL* (Osband et al., 2013; Azar et al., 2017; Ouhamma et al., 2023), the agent interacts with the environment in multiple episodes of $H$ steps. An episode starts by observing state $s_1$, then for $t = 1, \ldots, H$, the agent draws action $a_t$ from a (possibly time-dependent) policy $\pi_t(\cdot \mid s_t)$, observes the reward $r(s_t, a_t)$, and transits to a state $s_{t+1} \sim T(\cdot \mid s_t, a_t)$. Here, the value function and the state-action value functions at step $h \in [H]$ are respectively defined as $V_h^\pi(s) \triangleq \mathbb{E}_\pi \left[ \sum_{t=h}^{H} r(s_t, a_t) \mid s_h = s \right]$ and $Q_h^\pi(s, a) \triangleq \mathbb{E}_\pi \left[ \sum_{t=h}^{H} r(s_t, a_t) \mid s_h = s, a_h = a \right]$. Following (Altman, 1999),

we can define a finite-horizon version of occupancy measures as

$$v_\pi^H(s, a) \triangleq \frac{1}{H} \sum_{t=1}^{H} \mathbb{P}(s_t = s, a_t = a \mid \pi, \mu). \tag{7}$$

Following Syed et al. (2008), work by Kalagarla et al. (2021) shows that $v_\pi^H$ can be used in the linear programming formulation for solving MDPs and satisfies the *Bellman Flow Constraints* (in Equation 19 from Appendix A.3). We prove that under some assumptions, the finite-horizon occupancy measures also construct a manifold, referred to as $\mathcal{M}^H$.

**Proposition 3** (Properties of $\mathcal{M}^H$). *If the policy $\pi$ has a smooth parametrization $\theta$, then the space of finite-horizon occupancy measures $\mathcal{M}^H$ is a differentiable manifold. (Proof in Appendix A.5)*

This allows us to similarly define a Wasserstein metric on this manifold, which in turn, allows us to compute ESL and OMR for evaluating different RL algorithms.

## 4 Computational Challenges and Solutions

Similar to regret, our method requires knowing the optimal policy. This is because the efficiency and effectiveness of exploratory processes of RL algorithms are highly coupled with their ability to reach optimal policy. ESL and OMR depend on the policies being stationary and Markovian.

### 4.1 Policy datasets for computing occupancy measures

We consider approximations of occupancy measures using datasets assumed to be drawn from these measures. We estimate the Wasserstein distance between the occupancy measures using a method introduced by Alvarez-Melis & Fusi (2020) known as the *optimal transport dataset distance* (OTDD). OTDD uses datasets to estimate the Wasserstein distance between the underlying distributions. See Appendix A.6 for a detailed account of OTDD.

**Definition 4** (Policy dataset). *A dataset of a policy $\mathcal{D}_\pi$ is a set of state-action pairs drawn from the policy's occupancy measure, i.e. $\mathcal{D}_\pi = \{(s_{(i)}, a_{(i)})\}_{i=1}^{m} \sim v_\pi$. These can be constituted from the rollouts generated by the policy during task execution.*

We know from imitation learning that if we are given $\mathcal{D}_\pi$, generated by an expert policy, we can train a policy model on it in a supervised manner via behaviour cloning (Hussein et al., 2017). Thus, knowing $\mathcal{D}_\pi$ can allow converting an RL task into a Supervised Learning (SL) task. Consider a scenario when we have access to a sequence of datasets $(\mathcal{D}_{\pi_0}, \dots, \mathcal{D}_{\pi_N})$, each corresponding to policy $\pi_t$ for $t \geq 0$. If we train (in a supervised manner) a policy model sequentially on these datasets, the model will undergo a similar policy evolution as the RL algorithm that generated the policy trajectory $(\pi_t)_{t \geq 0}$. This allows us to conceptualise learning in RL as a sequence of SL tasks with sequential transfer learning across the datasets $(\mathcal{D}_{\pi_0}, \dots, \mathcal{D}_{\pi_N})$. We employ OTDD to estimate $\mathcal{W}_1(v_{\pi_k}, v_{\pi_{k+1}})$ using these datasets, i.e. $d_{OT}(\mathcal{D}_{\pi_k}, \mathcal{D}_{\pi_{k+1}}) \approx \mathcal{W}_1(v_{\pi_k}, v_{\pi_{k+1}})$, based on Proposition 4.

**Proposition 4** (Upper Bound on Estimation Error). *Let an RL algorithm yield a sequence of policies $\pi_0, \dots, \pi_N$ while training. Now, we construct $N$ datasets $\mathcal{D}_{\pi_0}, \dots, \mathcal{D}_{\pi_N}$, each consisting of $M$ rollouts of the corresponding policies. Then, we can use these datasets to approximate $\sum_{k=0}^{N-1} \mathcal{W}_1(v_{\pi_{\pi_k}}, v_{\pi_{\pi_{k+1}}})$ by $\sum_{k=0}^{N-1} d_{OT}(\mathcal{D}_{\pi_k}, \mathcal{D}_{\pi_{k+1}})$ with an expected error upper bound $\frac{2N\mathcal{E}_2}{\sqrt{M}} + N\gamma^{T+1} diam(\mathcal{SA})$. Here, $T$ is the total number of steps per episode, $diam(\mathcal{SA})$ is the diameter of the state-action space, and $\mathcal{E}_2$ is a positive-valued and polylogarithmic function of $S$ and $A$. For finite horizon case, we can further reduce the error bound to $\frac{2N\mathcal{E}_2}{\sqrt{M}}$.*

Proof of Proposition 4 is in Appendix A.7. The results support that ESL and OMR can be estimated as

$$\bar{\eta} = \mathbb{E}_{\pi_0, \mu} \left[ \frac{\sum_{k=0}^{N-1} d_{OT}(\mathcal{D}_{\pi_k}, \mathcal{D}_{\pi_{k+1}})}{d_{OT}(\mathcal{D}_{\pi_0}, \mathcal{D}_{\pi_N})} \right], \quad \text{and} \quad \bar{\kappa} = \mathbb{E}_{\pi_0, \mu} \left[ \frac{\sum_{k \in K^+} d_{OT}(\mathcal{D}_{\pi_k}, \mathcal{D}_{\pi_{k+1}})}{\sum_{k=0}^{N-1} d_{OT}(\mathcal{D}_{\pi_k}, \mathcal{D}_{\pi_{k+1}})} \right]. \tag{8}$$

### 4.2 When an optimal policy is not reached

So far we have assumed that the algorithms converge at the optimal policy, i.e. $\pi_N = \pi^*$. However, this is not always true. We consider a scenario when $\pi_N \neq \pi^*$, and define

$$\eta_{sub} = \frac{\sum_{k=0}^{N-1} \mathcal{W}_1(v_{\pi_{\pi_k}}, v_{\pi_{\pi_{k+1}}})}{\mathcal{W}_1(v_{\pi_0}, v_{\pi_N})}, \pi_N \neq \pi^* . \tag{9}$$

**Proposition 5.** *Given $N \geq 2$ and $\pi_0 \neq \pi_N \neq \pi^*$, we obtain*

$$\frac{\eta - \eta_{sub}}{\eta} \leq \frac{2\mathcal{W}_1(v_{\pi_N}, v_{\pi^*})}{\mathcal{W}_1(v_{\pi_0}, v_{\pi_N})} . \tag{10}$$

This is true due to the triangle inequalities: $\mathcal{W}_1(v_{\pi_0}, v_{\pi^*}) + \mathcal{W}_1(v_{\pi_N}, v_{\pi^*}) \geq \mathcal{W}_1(v_{\pi_0}, v_{\pi_N})$ and $\mathcal{W}_1(v_{\pi_{N-1}}, v_{\pi_N}) + \mathcal{W}_1(v_{\pi_N}, v_{\pi^*}) \geq \mathcal{W}_1(v_{\pi_{N-1}}, v_{\pi^*})$. The proof is provided in Appendix A.8. Note that Equation (10) shows that when $\pi_N$ is close to $\pi^*$, then $\eta_{sub}$ is a good approximation of $\eta$, and thus a good quantifier to determine the efficiency of the algorithm's exploratory process. However, $\mathcal{W}_1(v_{\pi_N}, v_{\pi^*})$ is dependent on the RL algorithm and hence a bound cannot be provided here. Still, $\eta_{sub}$ might be useful when $\pi_N$ is closer to $\pi^*$ than $\pi_0$. A fallible proxy for this could be when the performance of $\pi_N$ is better than that of $\pi_0$, i.e. $J_\mu^{\pi_N} > J_\mu^{\pi_0}$. We show the usefulness of $\eta_{sub}$ in our experimental results in Section 5.3 and Appendix B.5 for simple environments. It remains to be seen how useful $\eta_{sub}$ is in complex environments.

## 5 Experimental Evaluation

In this section, we evaluate the proposed methods in the *2D-Gridworld* and *Mountain Car* (Moore, 1990; Brockman et al., 2016) environments, to analyze our methods in discrete and continuous state-action spaces respectively. The 2D-Gridworld environment is of size 5×5 with actions: {up, right, down, left}. In the gridworld, we perform experiments on 3 settings namely:- A) deterministic with dense rewards, B) deterministic with sparse rewards, and C) stochastic with dense rewards. Further details about these settings are provided in Appendix B.1. The Mountain Car environment, in our experimentation, is a deterministic MDP with dense rewards that consists of both continuous states and actions - described in detail in (Brockman et al., 2016). Note that we used $L1$ distance and $L2$ distance as metrics ($d_\mathcal{X}$) for the state spaces of the 2D-Gridworld and Mountain Car, respectively, which underpin the $\mathcal{W}_1$. Our code is available at: `https://github.com/nkhumise-rea/analysis_of_occupancy_measure_trajectory`.

Our experiments aim to address the following questions:
1. *What information can the visualization of the policy evolution during RL training provide about the exploratory process of the algorithm?*
2. *How do ESL and OMR allow us to analyze the exploratory processes of RL algorithms?*
3. *Does ESL scale proportionally with task difficulty?*

**Summary of Results.** In Section 5.1, we demonstrate that visualizing the evolution of *distance-to-optimal* and *stepwise-distance* of different RL algorithms during training reveals: 1) whether the agent is stuck in suboptimal policies, 2) the coverage area of the exploration processes, and 3) their varied characteristics over time. We further compare ESL and OMR of different algorithms on a few environments in Section 5.2. Finally, we show in Section 5.4 that ESL scales proportionally with task difficulty, and thus, reflects the effects of task difficulty on exploration and learning.

### 5.1 Exploration Trajectories of RL Algorithms

(I) DISCRETE MDP. To understand the utility of visualizing exploratory processes, we use the following RL algorithms: 1) Tabular Q-learning with a) $\epsilon$-greedy ($\epsilon = 0$) and b) $\epsilon$-greedy ($\epsilon = 1$) strategies; 2) UCRL2 (Jaksch et al., 2010); 3) PSRL (Osband et al., 2013); 4) SAC (Haarnoja et al., 2018; Christodoulou, 2019); and 5) DQN (Mnih et al., 2013) with $\epsilon$-decay. The algorithms solve a simple 5×5 gridworld with dense rewards, starting from top-left (0,0) to reach bottom-right (4,4). Figure 3 presents exploratory behaviours of the algorithms in both the occupancy measure space and state space.

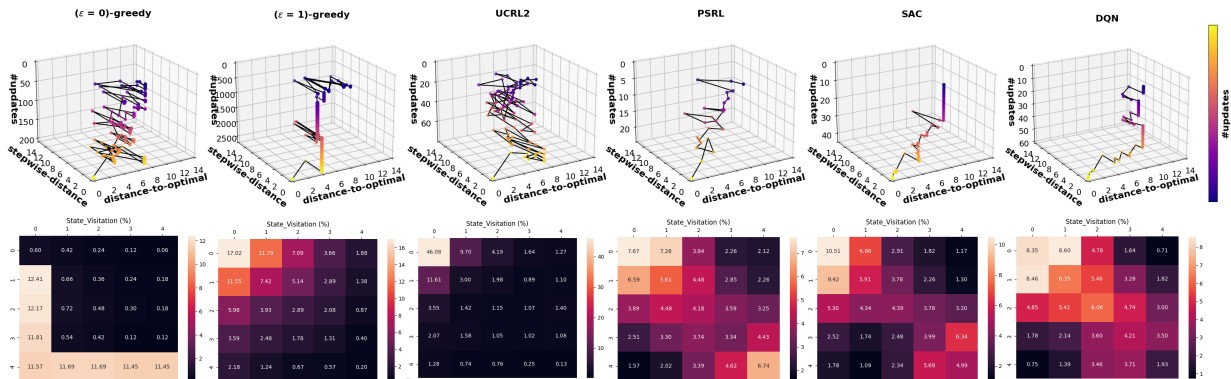

Figure 3: Top row: 3D plots of *distance-to-optimal* (x-axis) and *stepwise-distance* (y-axis) across number of updates (z-axis), illustrating policy evolution in the occupancy measure space for algorithms: $\epsilon(=0)$-greedy and $\epsilon(=1)$-greedy Q-learning, UCRL2, PSRL, SAC, and DQN (left to right). Bottom row: Corresponding state visitation frequencies over the full training. The problem setting is deterministic with dense rewards and 15 maximum number of steps per episode. (NB. Larger versions of these plots are presented in Appendix D.1, while their 2D projections are in Figure 15 at Appendix D.3, and corresponding performance plots are in Figure 12 at Appendix D.2.)

**Q-learning: $\epsilon = 0$ vs $\epsilon = 1$.** Note that $\epsilon = 0$ updates the Q-table by only exploiting, while $\epsilon = 1$ by exploring. From the state visitations, we observe expected characteristics, like a preferred visit path for $\epsilon = 0$, versus $\epsilon = 1$ with visitation frequencies that are similar at states equidistant from the start-state and gradually decreasing as the distance from the start-state increases. From the policy evolution, we see how scattered and erratic the policy transitions are for $\epsilon = 0$. Whereas $\epsilon = 1$ is dominated by unchanging or little-changing policies seen by straight vertical line segments (indicating being *stuck in suboptimality*). In this setting, $\epsilon = 0$ is characterised by transitioning between diverse policies (i.e. being aggressive with larger coverage area) while $\epsilon = 1$ is likely to be stuck in suboptimality. The stuckness is due to high action randomness in $\epsilon = 1$ that causes the agent to select suboptimal actions, slowing the convergence of the Q-table and not changing the learning policy until the best actions are discovered.

**UCRL2 vs PSRL.** UCRL2 has nearly uniform state visits (with the exception of the start-state because the initial state distribution is 1 at state (0,0)), thus being consistent with literature since the algorithm selects exploratory state-action pairs more uniformly (Jaksch et al., 2010). In contrast, PSRL has high visit frequencies along the diagonal states, because it selects actions according to the probability that they are optimal (Osband et al., 2013). We observe from the policy evolution plots that PSRL has smoother policy transitions that are orientated towards optimality, while UCRL2 behaves more aggressively with policy transitions that do not taper as it approaches optimality. Osband et al. (2013) highlighted that exploration in PSRL is guided by the variance of sampled policies as opposed to optimism in UCRL2. We observe in Figure 3 that the guiding variance in PSRL reduces after every policy update until optimality is reached, while UCRL2 maintains high variance.

**SAC vs DQN.** The state visits of both the algorithms appear to be similar. SAC has higher visitation frequencies at the corners than DQN. Surprisingly from the policy evolution plots, we learn that both algorithms have a reluctance to transition between policies - hence the *stuck in suboptimality* vertical line segments, especially initially. This reluctance is due to the slow 'soft updates' of target networks (Lillicrap et al., 2016) in the algorithms. We also observe that SAC approaches optimality more gradually than DQN.

**All algorithms.** Figure 3 shows that UCRL2 was more meandering (with larger coverage area) towards optimality than the rest. SAC and DQN approached optimality more directly and smoothly (with smaller coverage area) than the rest. These characteristics are intuitively revealed by policy visualisation plots, and are aligned with literature, hence enhancing our understanding of the exploratory processes.

(II) CONTINUOUS MDP. We use DDPG (Lillicrap et al., 2016) and SAC to solve the Mountain Car. The policy evolutions of these algorithms are presented in Figure 4.

**DDPG vs SAC.** Both exhibit short-distances $(< 1)$ between policy updates (i.e. small coverage area). They depict no sign of being stuck or settling early on any particular policy, which shows their continuously exploratory nature. While they begin with almost constant mean *distances-to-optimal* and *stepwise-distances*, SAC drops its mean *distance-to-optimal* earlier than DDPG.

Figure 4 illustrates how OMR changes with update number $k$. OMR($k$) represents OMR starting with the $k^{\text{th}}$ policy as the initial policy, while OMR starts from the $0^{\text{th}}$ policy (details of computing OMR($k$) are in Appendix B.2). For both algorithms, OMR($k$) remains near chance level ($\sim 0.5$) initially, then sharply increases near the final updates. This suggests that early policy updates are purely exploratory and oblivious to policy improvement but align with the optimal policy just before convergence. The efficiency of the algorithm depends on how early this transition occurs, e.g. starts earlier for SAC than DDPG, rendering SAC more efficient.

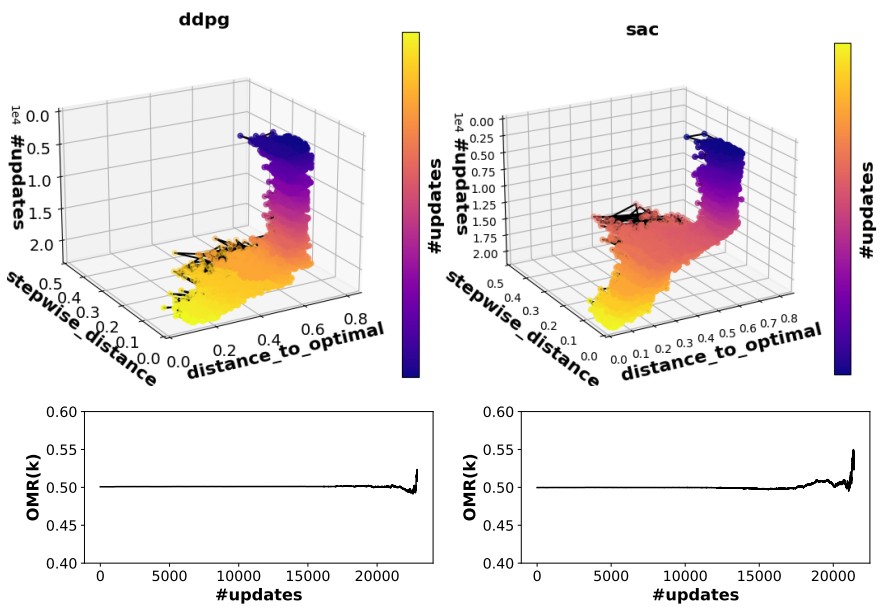

Figure 4: Top row: 3D plots of *distance-to-optimal* and *stepwise-distance* vs. number of updates for DDPG and SAC. Bottom row: OMR($k$) vs. #update, $k$, for the corresponding algorithms. Note that corresponding performance plots are in Figure 13 in Appendix D.2.

## 5.2 Comparison of ESL and OMR across RL Algorithms and Environments, and their complementarity to number of updates and regret

Tables 1-3 showcase how ESL and OMR are summary indices of the policy trajectories during learning by evaluating the algorithms in various settings.

**Dense Rewards.** We observe, in Table 1, that PSRL took the lowest number of updates (UC) to reach the optimal policy in contrast with SAC. Yet, PSRL was meandering more than SAC (see Figure 3). The relative directness of SAC is captured by lower ESL and higher OMR compared to PSRL. Even though SAC has larger UC than PSRL, it took a shorter path to optimality than PRSL. This shows that UC does not necessary correlate with ESL and OMR, and it provides incomplete information about the exploratory processes. Indeed, two algorithms may have the same UC, but different ESL and/or OMR due to different *step-wise distances* and varied movement towards optimality.

**Sparse Rewards.** In the sparse rewards setting (Table 2), low performance of DQN is observed in both our indices and UC. However, SAC is more efficient with lowest ESL and highest OMR, yet UCRL2 has the lowest number of updates (UC). Note that *UCRL2 is provably regret-optimal, while SAC does not have such rigorous theoretical guarantees but is known to be practically efficient, and this is well captured by ESL and*

| Algo. | ESL | OMR | UC | SR% |
|---|---|---|---|---|
| SAC | 9.26±5.54 | 0.58±0.14 | 980±670 | 100 |
| UCRL2 | 47.2±8.20* | 0.49±0.04 | 60.7±11 | 100 |
| PSRL | 23.2±11.5 | 0.52±0.06 | **34.1±9.34** | 100 |
| DQN | 12.4±7.13 | 0.54±0.11 | 161±93 | 98 |
| $\epsilon$(=1)-greedy | **6.27±2.22** | **0.61±0.09** | 672±385 | 100 |
| $\epsilon$(=0.9)-decay | 8.10±3.43 | 0.61±0.10 | 389±138 | 100 |
| $\epsilon$(=0)-greedy | 15.5±5.28 | 0.53±0.06 | 176±37.9 | 84 |

Table 1: Evaluation of RL algorithms (over 40 runs) in the **deterministic, dense-rewards setting** for 5×5 gridworld, including Effort of Sequential Learning (ESL), Optimal Movement Ratio (OMR), number of updates to convergence (UC), and success rate (SR). Lowest ESL, lowest UC, and highest OMR values are in **bold**. The highest ESL value is starred (*).

| Algo. | ESL | OMR | UC | SR% |
|---|---|---|---|---|
| | **Deterministic, sparse** | | | |
| SAC | **27.8±21.9** | **0.57±0.13** | 4385±3274 | 100 |
| UCRL2 | 73.3±0.0 | 0.45±0.0 | **93.0±0.0** | 100 |
| PSRL | 73.2±54.1 | 0.52±0.076 | 100±67.3 | 100 |
| DQN | 137±154* | 0.49±0.08 | 12638±4431 | 80 |
| | **Stochastic, dense** | | | |
| SAC | 445±245 | 0.501±0.004 | 2463±2043 | **92** |
| UCRL2 | 198±121 | 0.502±0.027 | 268±155 | 32 |
| PSRL | **55.4±33.6** | **0.52±0.04** | **76.1±50.6** | **92** |
| DQN | 458±311* | 0.502±0.01 | 1586±1077 | 24 |

Table 2: Evaluation of RL algorithms (over 40 runs) in the **deterministic, sparse-rewards** and **stochastic, dense-rewards** settings for 5×5 gridworld. Lowest ESL, highest OMR and lowest UC values are in **bold**. The highest ESLs are starred.

*OMR.* So far, Tables 1 and 2 demonstrate how our indices provide a clearer picture of exploratory processes than the number of updates.

**Stochastic Transitions.** In the stochastic setting (Table 2), by observing only successful cases, we notice that the meandering characteristic of PSRL and UCRL2 is more suitable for this setting than SAC and DQN (based on better ESL and OMR values). PSRL and UCRL2 have similar regret bounds (Osband et al., 2013); yet in Tables 1 and 2, PSRL has better ESL and OMR (along with a higher success rate). This aligns with the regret analysis presented in (Osband et al., 2013).

Table 3 corroborates with the policy evolution plots in Figure 4, in that due to SAC dropping its mean *distance-to-optimal* earlier than DDPG it exhibits a lower ESL. Additionally, we notice a trend of increasing ESL and decreasing OMR across algorithms when shifting from dense-rewards to sparse-rewards settings, from deterministic to stochastic transitions, from discrete to continuous environments, indicating an increase in the effort of the exploratory processes. We have shown that ESL and OMR enhance the understanding of exploratory processes by effectively summarizing the policy trajectories of algorithms during learning. They offer more insight than the number of updates, and align with regret when algorithms reach an optimal policy. In the next Section, we demonstrate the utility of ESL when an optimal policy is not reached.

| Algo. | ESL | OMR | UC | SR% |
|---|---|---|---|---|
| DDPG | 1881±500 | 0.501 | 23500±5268 | 100 |
| SAC | 1619±189 | 0.5 | 22700±2971 | 100 |

Table 3: Evaluation of RL algorithms in the Mountain Car continuous MDP (over 5 runs). The variances for OMR are negligible.

## 5.3 Usefulness of ESL when optimal policy is not reached

When an optimal policy is not reached at the end of learning, ESL cannot be computed exactly. At this point, we propose to use an approximation of ESL, i.e. $\eta_{sub}$ (Equation 9). A natural question arises: *When the optimal policy is not reached, does the $\eta_{sub}$ still yield insights about the exploratory process of RL?* In Table 4, we compare ESL when the optimal policy was reached, i.e. $\eta$, versus when it was not, i.e. $\eta_{sub}$.

We observe that $\eta_{sub}$ values are always greater than $\eta$ values. However, they both yield the same efficiency ranking (e.g. PSRL, UCRL2, SAC and DQN). This indicates that $\eta_{sub}$ reliably predicts results provided by $\eta$ for relative comparison of algorithms.

| Algo. | $\eta$ | $\eta_{sub}$ | d | c |
|-------|--------|--------------|---|---|
| SAC | 445±246 | 853±127 | 5.63±1.23 | 7.26±1.45 |
| UCRL2 | 198±121 | 510±274 | 5.36±0.84 | 4.58±1.90 |
| PSRL | 55.4±33.6 | 361±43.6 | 4.97±1.34 | 3.91±0.48 |
| DQN | 458±311 | 1971±250 | 4.88±1.06 | 6.52±0.31 |

Table 4: Evaluation of algorithms in the **stochastic, dense-rewards setting** for 5×5 gridworld. When the algorithm converged to optimality, $\eta$ is the Effort of Sequential Learning, and $d = \mathcal{W}_1(v_{\pi_0}, v_{\pi^*})$ is the distance between the initial and optimal policies. When the algorithm did not converge to the optimal policy but some $\pi_N$, we used $\eta_{sub}$ and $c = \mathcal{W}_1(v_{\pi_0}, v_{\pi_N})$ to denote the aforementioned quantities. 40 training trials were used.

### 5.4 ESL Increases with Task Difficulty

Figure 5 depicts the ESL values for Q-learning with $\epsilon$-decay strategy (for $\epsilon = 0.9$) across tasks with varying hardness. These tasks are deterministic 2D-Gridworld of sizes 5×5 and 15×15 matched with either dense or sparse rewards (as specified in Appendix B.1). We chose to assess the $\epsilon$-decay Q-learning algorithm because it is simple and yet completes all these tasks. We observe that the ESL is lowest for *[5×5] dense* (5×5 grid, dense rewards) and highest for *[15×15] sparse* (15×15 grid, sparse rewards) as anticipated. The results demonstrate that ESL scales proportionally with task difficulty, matching expectations that more difficult tasks demand greater effort of the exploratory process.

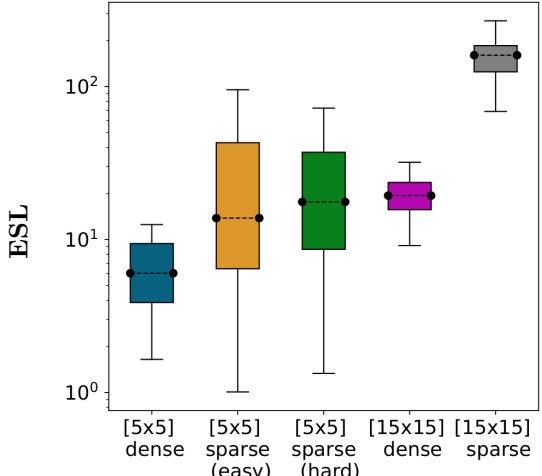

Figure 5: Q-learning with $\epsilon$-greedy ($\epsilon = 0.9$ decaying, averaged over 40 runs) across deterministic 2D-Gridworld (5×5 and 15×15) tasks. The 1st and 4th (from left to right) have dense rewards, while the rest have sparse rewards (details in Appendix B.1).

**Remarks.** When the optimal policy is reached, we can use the visualisation of policy trajectories, as well as ESL ($\eta$) and OMR to study exploratory processes of an RL algorithm. When the optimal policy is not reached, we can still use the visualisation of policy trajectories and ESL-sub ($\eta_{sub}$) to study exploratory processes because they still capture characteristics of exploratory processes (e.g. high coverage, smooth exploration). Additionally, ESL ($\eta$ or $\eta_{sub}$) captures the hardness of the task that we are solving. Thus, studying the occupancy measure trajectories and their corresponding indices can aid in making a knowledgeable choice of an RL algorithm that exhibits desired characteristics.

## 6 Related Works

Several prior works have utilized various components leveraged in our work, namely Wasserstein distance, occupancy measures, and the trajectory of RL on a manifold, but for different purposes. Here, we summarise them and elucidate the connections.

In supervised learning, Alvarez-Melis & Fusi (2020) proposed an optimal transport approach, namely Optimal Transport Dataset Distance (OTDD), to quantify the transferability between two supervised learning tasks by computing the similarity (i.e. distance) between the task datasets. Here, we conceptualise and define the effort of learning for RL, as a sequence of such supervised learning tasks. We observe that *the total effort of sequential learning can be computed as the sum of OTDD distances between consecutive occupancy measures.* Recently, Zhu et al. (2024) have developed generalized occupancy models by defining cumulative features that are transferable across tasks. In future, one can generalize our indices for the cumulative features constructed from some invertible functions of the step-wise occupancy measures.

Optimal transport-based approaches are also explored in RL literature. These works broadly belong to two families. First line of works uses Wasserstein distance over a posterior distribution of Q-values (Metelli et al., 2019; Likmeta et al., 2023) or return distributions (Sun et al., 2022) to quantify uncertainty, and then to use this Wasserstein distance as a loss to learn better models of the posterior distribution of Q-values or return distributions, respectively. The second line of works uses Wasserstein distance between a feasible family of MDPs as an additional robustness constraint to design robust RL algorithms (Abdullah et al., 2019; Derman & Mannor, 2020; Hou et al., 2020). Here, *we introduce a concept of using Wasserstein distance between occupancy measures to understand the exploratory dynamics.* Incorporating this insight into better algorithm design would be an interesting future work. Recently, Calo et al. (2024) related Wasserstein distance between reward-labelled Markov chains to bisimulation metrics which abstract state spaces. In the same spirit, we could use reward as the cost-function in computing our nested Wasserstein distance (OTDD) to obtain a reward- or value-aware OTDD to define broader bisimulation metrics with abstract state-action spaces, instead of just state spaces.

As a parallel approach to optimal transport, the information geometries of the trajectory of an RL algorithm under different settings are studied. These approaches use mutual information as a metric instead of Wasserstein distance. Basu et al. (2020) studied the information geometry of Bayesian multi-armed bandit algorithms. They considered a bandit algorithm as a trajectory on a belief-reward manifold, and proposed a geometric approach to design a near-optimal Bayesian bandit algorithm. Eysenbach et al. (2021); Laskin et al. (2022) studied information geometry of unsupervised RL and proposed mutual information maximisation schemes over a set of tasks and their marginal state distributions. Yang et al. (2024) extended this approach with Wasserstein distance and demonstrated benefits of using Wasserstein distance than mutual information. *We use Wasserstein distance as a natural metric in occupancy manifold which aligns with the hardness of different tasks.* It would be interesting to extend our framework to understand the dynamics of unsupervised RL algorithms.

## 7 Discussion

Our work introduces methods to theoretically and quantitatively understand and compare the exploratory strategies of different RL algorithms. Since learning in a typical RL algorithm happens through a sequence of policy updates, we propose to understand the exploratory process by visualizing and analyzing the path traversed by an RL algorithm in the space of occupancy measures. We show the usefulness of this approach by conducting experiments on various environments and RL algorithms.

Our results show that ESL and OMR provide insight into the evolution of the agent's policy, revealing whether it is approaching the optimal policy in a steady or meandering manner. Additionally, they allow us to understand how the learning process of the same algorithm changes with different rewards and transitions structures, and task hardness. We emphasize that ESL and OMR complement the number of updates to converge and regret (see Appendix B.7) rather than replace them.

We now discuss various practical aspects and future possibilities.

**Computational complexity.** While efficiency is an important aspect, the primary focus of this work is to introduce a framework for analyzing exploration in RL using occupancy measures. It is of interest to utilize our approach to benchmark and compare the learning dynamics of various RL algorithms in more environments, especially large-scale or high-dimensional ones. Computing Wasserstein distances in such environments would incur high computational costs, however in recent years several methods such as greedy computation (Carlier et al., 2010), hierarchical methods (Lee et al., 2019), and inexact proximal point methods (Xie et al., 2020) have been introduced to handle large-scale OT problems. For example, (Gao & Chaudhari, 2021) leverages a block-diagonal approximation method to deal with high-dimensional probability distributions similar to ours, while anchor space OT (Huang et al., 2024) specifically addresses multiple OT problems with multiple distributions.

**Use of 1-Wasserstein metric $\mathcal{W}_1$.** The $\mathcal{W}_1$ satisfies the Kantorovich-Rubinstein duality, making Equation (3) applicable and providing a basis for bounding regret in Proposition 2. In contrast, $\mathcal{W}_{p>1}$ does not have such duality. Since $\mathcal{W}_1(P,Q) \leq \mathcal{W}_{p>1}(P,Q)$ (Villani, 2009), $\mathcal{W}_1$ yields tighter regret bounds. Moreover, $\mathcal{W}_1$ is less sensitive to outliers and sampling discrepancies than $\mathcal{W}_{p>1}$ (Raghvendra et al., 2024), making it well-suited for our setting. Nevertheless, our approach extends to $\mathcal{W}_{p>1}$. Compared to KL-divergence, $\mathcal{W}_1$ is a metric that satisfies symmetry and triangle inequality, which have been instrumental in our proofs and guarantees, while KL-divergence is not a metric. Additionally, $\mathcal{W}_{p\geq1}$ leverages the geometry of the underlying support space (Peyré, 2019). This allows it to capture distances between distributions with disjoint support (Peyré, 2019), which KL-divergence cannot capture. Hence, we use $\mathcal{W}_1$ in our approach.

**Choice of distance metric $d_\mathcal{X}$.** The choice of the distance metric impacts the geometry of the support space (Lee, 2009), consequently the Wasserstein distances, and thus ESL and OMR. In our case, the support space (i.e. state-action space) is reduced to the state space, as the action space maps back to the state space via OTDD. Thus, the choice of distance metric should reflect the effort of moving in the state space. For example, in the Gridworld, we used L1 distance because only vertical and horizontal displacements are allowed, and L2 distance in the Mountain Car, as applicable to real-world spaces.

**Regret and *Sum-of-distances-to-optimal* ($\sum_{k=1}^N \mathcal{W}_1(v_{\pi_k}, v_{\pi^*})$).** Comparing regret across policies measures the similarity of returns (with respect to optimal), disregarding behavioural differences, like variations in actions at the same states. This makes regret advantageous in settings without critical safety and physical constraints, e.g. games, due to its computational efficiency. In contrast, the *sum-of-distance-to-optimal* focuses on behavioural differences between policies, is reward-agnostic, and is well-suited for environments where safety and physical constraints are critical, e.g. robotics. While minimizing regret prioritizes matching the performance of the optimal policy, minimizing the *sum-of-distance-to-optimal* focuses on mimicking its behaviour. Thus, the *sum-of-distance-to-optimal* can be used similarly to regret, especially where the behaviour of achieving good performance is essential.

**Algorithm Selection and Design.** Depending on the environment, we can select an algorithm with promising characteristics and spend more time optimising it to improve performance. For example, rather than tuning the hyper-parameters of multiple competing algorithms to find the best one, it may be more effective to first identify an algorithm best suited to an environment based on ESL and OMR, and then fine-tune it. The chosen algorithm might remain suitable across similar environments as well.

Furthermore, we could incorporate an online adaptation of the exploratory process of the RL algorithm itself, based on recent estimates of the ESL and OMR. For example, if the current policy gives a better return than the initial policy, then we could adapt the exploratory parameters (at a slow rate) to optimise a running estimate of $\eta_{sub}$ and a suitable approximation of OMR, thus enabling better exploration. However, the feasibility and convergence of such a scheme remain open.

**Acknowledgments**

R. Nkhumise expresses his gratitude to Pawel Pukowski and Mohamed S. Talamali for their insightful discussions and valuable feedback on the ideas explored in this study. R. Nkhumise was supported by the EPSRC Doctoral Training Partnership (DTP) - Early Career Researcher funding awarded to A. Gilra. A. Gilra acknowledges the CHIST-ERA grant for the Causal Explanations in Reinforcement Learning (CausalXRL) project (CHIST-ERA-19-XAI-002), by the Engineering and Physical Sciences Research Council, United

Kingdom (grant reference EP/V055720/1) for supporting the work. D. Basu acknowledges the CHIST-ERA grant for the CausalXRL project (CHIST-ERA-19-XAI-002) by L'Agence Nationale de la Recherche, France (grant reference ANR-21-CHR4-0007), the ANR JCJC for the REPUBLIC project (ANR-22-CE23-0003-01), and the PEPR project FOUNDRY (ANR23-PEIA-0003) for supporting the work. We thank Eleni Vasilaki and Philippe Preux for their support.

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

## A Theoretical Analysis

### A.1 MDP with Lipschitz Rewards

Given two metric spaces $(\mathcal{X}, d_{\mathcal{X}})$ and $(\mathcal{Y}, d_{\mathcal{Y}})$, a function $f : \mathcal{X} \to \mathcal{Y}$ is called 1-Lipschitz continuous if (Villani, 2009):

$$d_Y(f(x), f(x')) \le d_X(x, x'), \forall (x, x') \in X \tag{11}$$

This implies that the Lipschitz semi-norm over the function space $\mathcal{F}(X, Y)$, defined as

$$\|f\|_L = \sup_{x \neq x'} \left\{ \frac{d_Y(f(x), f(x'))}{d_X(x, x')} \mid \forall (x, x') \in \mathcal{X} \right\}, \tag{12}$$

is $\le 1$. When $(\mathcal{X}, d_{\mathcal{X}})$ is a Polish space and $\mu, \nu \in \mathcal{P}(\mathcal{X})$, the **Kantorovich-Rubinstein** formula states that (Villani, 2009):

$$\begin{aligned} \mathcal{W}_1(\mu, \nu) &= \sup_{\|f\|_L \le 1} \left\{ \int_{\mathcal{X}} f \, d\mu - \int_{\mathcal{X}} f \, d\nu \right\} \\ &= \sup_{\|f\|_L \le 1} \left\{ \mathbb{E}_\mu \left[ f(X) \right] - \mathbb{E}_\nu \left[ f(X) \right] \right\}, \end{aligned} \tag{13}$$

where $\mathcal{W}_1(\mu, \nu)$ is the 1-Wasserstein distance between $\mu$ and $\nu$ with $f$ as the cost function.

Note that when $\|f\|_L \le L_{\mathcal{R}}$ for any $L_{\mathcal{R}} > 0$, then the function $f$ is called $L_{\mathcal{R}}$-Lipschitz continuous, and Equation 13 becomes (Gelada et al., 2019),

$$\mathcal{W}_1(\mu, \nu) = \frac{1}{L_{\mathcal{R}}} \sup_{\|f\|_L \le L_{\mathcal{R}}} \left\{ \mathbb{E}_\mu \left[ f(X) \right] - \mathbb{E}_\nu \left[ f(X) \right] \right\}. \tag{14}$$

Now, we consider $\mathcal{X} = \mathcal{S} \times \mathcal{A}$, i.e. the state-action space, $\mathcal{Y} = \mathbb{R}$, i.e. the real line, and the function $f$ to be the reward function $\bar{\mathcal{R}}$. Then, we can call the reward function $\bar{\mathcal{R}}$ to be $L_{\mathcal{R}}$-Lipschitz if

$$|\bar{\mathcal{R}}(s, a) - \bar{\mathcal{R}}(s', a')| \le L_{\mathcal{R}} d_{\mathcal{S}\mathcal{A}}((s, a), (s', a'))$$

for all $s, s' \in \mathcal{S}$, and $a, a' \in \mathcal{A}$, and $d_{\mathcal{S}\mathcal{A}}((s, a), (s', a')) = d_{\mathcal{S}}((s, s')) + d_{\mathcal{A}}((a, a'))$ being the metric on the state-action space $\mathcal{S} \times \mathcal{A}$. If the reward function $\bar{\mathcal{R}}$ of an MDP is $L_{\mathcal{R}}$-Lipschitz, we refer it as an MDP with Lipschitz rewards.

### A.2 Performance Difference and Occupancy Measures

We know that

$$J_\mu^\pi = \frac{1}{\rho} \mathbb{E}_{(s,a) \sim v_\pi} \left[ \bar{\mathcal{R}}(s, a) \right] . \tag{15}$$

Using Equation 15, we write for two policies $\pi$ and $\pi'$, with $\mu(s)$ as the initial state distribution,

$$\left| J_\mu^\pi - J_\mu^{\pi'} \right| = \frac{1}{\rho} \left| \mathbb{E}_{(s,a) \sim v_\pi} \left[ \bar{\mathcal{R}}(s, a) \right] - \mathbb{E}_{(s,a) \sim v_{\pi'}} \left[ \bar{\mathcal{R}}(s, a) \right] \right| \tag{16}$$

Given an MDP with $L_{\mathcal{R}}$-Lipschitz rewards, the **Kantorovich-Rubinstein** formula dictates that (Gelada et al., 2019):

$$\sup_{\|\bar{R}\|_L \le L_{\mathcal{R}}} \left| \mathbb{E}_{(s,a) \sim v_\pi} \left[ \bar{\mathcal{R}}(s, a) \right] - \mathbb{E}_{(s,a) \sim v_{\pi'}} \left[ \bar{\mathcal{R}}(s, a) \right] \right| = L_{\mathcal{R}} \mathcal{W}_1(v_\pi, v_{\pi'}) \tag{17}$$

By dividing both sides of Equation 17 by $\rho$, and due to an upper bound by the supremum, this inequality follows:

$$\left| J_\mu^\pi - J_\mu^{\pi'} \right| \le \frac{L_{\mathcal{R}}}{\rho} \mathcal{W}_1(v_\pi, v_{\pi'}) \tag{18}$$

## A.3  Proof of proposition 1

The Linear Programming formulation for solving MDPs, assuming discrete state and action spaces, is (Puterman, 1994):

$$
\begin{aligned}
&\max_{v_\pi} \sum_{s,a} r(s,a) v_\pi(s,a) \\
&\text{subject to} \sum_{a} v_\pi(s,a) = p_0(s) + \gamma \sum_{s',a} T(s \mid s', a) v_\pi(s', a) \\
&v_\pi(s,a) \geq 0 \quad \forall (s,a) \in \mathcal{S} \times \mathcal{A},
\end{aligned}
\tag{19}
$$

where $p_0(s)$ is the initial state distribution and $T(s \mid s', a)$ is the transition probability. The constraints of this optimization problem are often referred to as *Bellman Flow Constraint*.

A stationary policy $\pi$ has a corresponding occupancy measure $v_\pi(s,a)$ that satisfies the Bellman flow constraint (Syed et al., 2008), and hence $\pi$ and $v_\pi(s,a)$ share a bijective relationship (Syed et al., 2008; Givchi, 2021),

$$
\pi(a \mid s) = \frac{v_\pi(s,a)}{u_\pi(s)}
\tag{20}
$$

with

$$
u_\pi(s) = \sum_{a'} v_\pi(s, a') = p_0(s) + \gamma \sum_{s',a'} T(s \mid s', a') v_\pi(s', a')
\tag{21}
$$

By rearranging Equation 20 to

$$
v_\pi(s,a) = \pi(a \mid s) u_\pi(s)
\tag{22}
$$

and substituting Equation 22 into Equation 21, we can rewrite Equation 21 as (defining $\mathcal{P}^\pi \triangleq \sum_a T(s \mid s', a) \pi(a \mid s')$),

$$
\begin{aligned}
p_0(s) &= u_\pi(s) - \gamma \sum_{s',a} T(s \mid s', a) \pi(a \mid s') u_\pi(s') \\
&\triangleq u_\pi(s) - \gamma \sum_{s'} \mathcal{P}^\pi(s \mid s') u_\pi(s')
\end{aligned}
\tag{23}
$$

which in matrix form is

$$
\begin{aligned}
\mathbf{p}_0 &= \mathbf{u}_\pi - \gamma \mathbf{P}^\pi \mathbf{u}_\pi \\
&= \left( \mathbb{I} - \gamma \mathbf{P}^\pi \right) \mathbf{u}_\pi,
\end{aligned}
\tag{24}
$$

where $\mathbf{p}_0, \mathbf{u}_\pi \in \mathbb{R}^{|\mathcal{S}|}$ are column vectors and $\mathbf{P}^\pi \in \mathbb{R}^{|\mathcal{S}| \times |\mathcal{S}|}$ are matrices. Solving for $\mathbf{u}_\pi$, we get

$$
\mathbf{u}_\pi = \left( \mathbb{I} - \gamma \mathbf{P}^\pi \right)^{-1} \mathbf{p}_0
\tag{25}
$$

The inverse matrix $\left( \mathbb{I} - \gamma \mathbf{P}^\pi \right)^{-1}$ exists because for $\gamma < 1$, $\left( \mathbb{I} - \gamma \mathbf{P}^\pi \right)$ is a strictly diagonally dominant matrix (Syed et al., 2008). Thus, $\left( \mathbb{I} - \gamma \mathbf{P}^\pi \right)^{-1} = \sum_{t=0}^{\infty} (\gamma \mathbf{P}^\pi)^t$, where $\sum_{t=0}^{\infty} (\gamma \mathbf{P}^\pi)^t$ forms a valid *Neumann series* (Ward, 2021). We let $\mathbf{A}^\pi = \sum_{t=0}^{\infty} (\gamma \mathbf{P}^\pi)^t$, so Equation 25 can be written as $\mathbf{u}_\pi = \mathbf{A}^\pi \mathbf{p}_0$. We can therefore express Equation 22 in matrix form as:

$$
\begin{aligned}
\mathbf{v}_\pi &= \mathbf{\Pi} \odot \left( \mathbf{u}_\pi^T \otimes \mathbf{1} \right)^T \\
&= \mathbf{\Pi} \odot \left( \mathbf{p}_0^T (\mathbf{A}^\pi)^T \otimes \mathbf{1} \right)^T,
\end{aligned}
\tag{26}
$$

where $\mathbf{\Pi}, \mathbf{v}_\pi \in \mathbb{R}^{|\mathcal{S}| \times |\mathcal{A}|}$, $\mathbf{1} \in \mathbb{R}^{|\mathcal{A}|}$ is a column vector of ones, $\otimes$ presents the Kronecker product, and $\odot$ denotes the Hadamard product.

If we consider the case of a parameterized policy $\mathbf{\Pi}(\theta)$, then the derivative of $\mathbf{v}_\pi$ with respect to $\theta$ is

$$
\begin{aligned}
\nabla_\theta \mathbf{v}_\pi &= \nabla_\theta \left[ \mathbf{\Pi} \odot \left( \mathbf{p}_0^T (\mathbf{A}^\pi)^T \otimes \mathbf{1} \right)^T \right] \\
&= \nabla_\theta \mathbf{\Pi} \odot \left( \mathbf{p}_0^T (\mathbf{A}^\pi)^T \otimes \mathbf{1} \right)^T + \mathbf{\Pi} \odot \nabla_\theta \left( \mathbf{p}_0^T (\mathbf{A}^\pi)^T \otimes \mathbf{1} \right)^T \\
&= \nabla_\theta \mathbf{\Pi} \odot \left( \mathbf{p}_0^T (\mathbf{A}^\pi)^T \otimes \mathbf{1} \right)^T + \mathbf{\Pi} \odot \left( \mathbf{p}_0^T (\nabla_\theta \mathbf{A}^\pi)^T \otimes \mathbf{1} \right)^T
\end{aligned}
\tag{27}
$$

The first term in Equation 27 is differentiable since the policy is parameterized by $\theta$. We expand $\nabla_\theta \mathbf{A}^\pi$ as follows:

$$
\begin{aligned}
\nabla_\theta \mathbf{A}^\pi &= \nabla_\theta \left( \sum_{t=0}^\infty (\gamma \mathbf{P}^\pi)^t \right) \\
&= \nabla_\theta \left( \mathbb{I} + (\gamma \mathbf{P}^\pi) + (\gamma \mathbf{P}^\pi)^2 + \cdots + (\gamma \mathbf{P}^\pi)^t + \ldots \right) \\
&= 0 + \nabla_\theta (\gamma \mathbf{P}^\pi) + 2(\gamma \mathbf{P}^\pi) \nabla_\theta (\gamma \mathbf{P}^\pi) + 3(\gamma \mathbf{P}^\pi)^2 \nabla_\theta (\gamma \mathbf{P}^\pi) + \cdots + t(\gamma \mathbf{P}^\pi)^{t-1} \nabla_\theta (\gamma \mathbf{P}^\pi) + \ldots \\
&= \left( \mathbb{I} + 2(\gamma \mathbf{P}^\pi) + 3(\gamma \mathbf{P}^\pi)^2 + \cdots + t(\gamma \mathbf{P}^\pi)^{t-1} + \ldots \right) \nabla_\theta (\gamma \mathbf{P}^\pi) \\
&= \left( \sum_{t=0}^\infty (t+1)(\gamma \mathbf{P}^\pi)^t \right) \nabla_\theta (\gamma \mathbf{P}^\pi) \\
&= \sum_{t=0}^\infty (t+1)(\gamma \mathbf{P}^\pi)^t \gamma \nabla_\theta \mathbf{P}^\pi \\
&\equiv \sum_{t=0}^\infty (t+1)(\gamma \mathbf{P}^\pi)^t \gamma \nabla_\theta \left[ \sum_{s',a} T(s|s',a)\pi(a|s') \right] \\
&= \sum_{t=0}^\infty (t+1)(\gamma \mathbf{P}^\pi)^t \gamma \left[ \sum_{s',a} T(s|s',a) \nabla_\theta \pi(a|s') \right] .
\end{aligned}
\tag{28}
$$

Since Equation 28 shows that $\nabla_\theta \mathbf{A}^\pi$ is differentiable, $\nabla_\theta \mathbf{v}_\pi$ is also differentiable based on Equation 27. Proceeding similarly, given the same conditions, we see that all higher derivatives of $\mathbf{v}_\pi$ also exist with respect to $\theta$. Thus, the space of parametrized occupancy measures $v_\pi$ forms a differentiable manifold.

### A.4 Proof of proposition 2

Regret is a common metric for evaluating agents, that measures the total loss an agent incurs over policy updates by using its policy in lieu of the optimal one, defined as (Osband et al., 2013),

$$
\text{Regret} = \mathbb{E}_{s\sim\mu} \left[ \sum_k (V^*(s) - V_{\pi_k}(s)) \right]
\tag{29}
$$

where $V^* = V_{\pi^*}$ is the value function of the optimal policy $\pi^*$ while $V_{\pi_k}(s)$ is the value function of policy $\pi_k$, and $\mu$ is the initial state distribution.

Since $J_\mu^\pi = \mathbb{E}_{s\sim\mu}[V_\pi(s)]$, we can conclude from Equation 29 that

$$
\begin{aligned}
\text{Regret} &= \mathbb{E}_{s\sim\mu} \left[ \sum_k (V^*(s) - V_{\pi_k}(s)) \right] \\
&= \sum_k \left[ \mathbb{E}_{s\sim\mu}(V^*(s) - V_{\pi_k}(s)) \right] \\
&= \sum_k \left( J_\mu^{\pi^*} - J_\mu^{\pi_k} \right) \\
&= \sum_k \left| J_\mu^{\pi^*} - J_\mu^{\pi_k} \right| \\
&\leq \sum_k \frac{L_\mathcal{R}}{\rho} \mathcal{W}_1(v_{\pi^*}, v_{\pi_k})
\end{aligned}
\tag{30}
$$

The last inequality is due to Equation 18.

## A.5 Proof of proposition 3

Let us begin the proof by defining the visitation probability at any step $h \in [H]$ in an episode, following policy $\pi(a|s)$. Specifically,

$$q_\pi^h(s,a) \triangleq \mathbb{P}(s_h = s, a_h = a) \ \ \forall h \in [H] \quad \text{and} \quad q_\pi^h(s,a) \triangleq 0 \ \ \forall h \in \mathbb{N} \wedge h > H \,. \tag{31}$$

Thus, we rewrite Equation 7 as $v_\pi^H(s,a) = \frac{1}{H} \sum_{h=1}^H q_\pi^h(s,a)$.

Then, following (Kalagarla et al., 2021), we can write the Linear Programming formulation for solving episodic MDP $\mathbb{M}^H$ as

$$\max_{\{q_\pi^h\}_{h=1}^H} \sum_{h,s,a} r(s,a) q_\pi^h(s,a)$$

$$\text{subject to} \sum_a q_\pi^h(s,a) = \sum_{s',a} T(s \mid s', a) q_\pi^{h-1}(s', a) \quad \forall h \in [H] \wedge h > 1 \,,$$

$$q_\pi^1(s,a) = \pi(a|s)\mu(s) \,, \tag{32}$$

$$q_\pi^h(s,a) \geq 0 \qquad \forall h \in [H], (s,a) \in \mathcal{S} \times \mathcal{A} \,,$$

where $\mu(s)$ is the initial state distribution and $T(s \mid s', a)$ is the transition probability. The constraints of this optimization problem are often referred to as *Bellman Flow Constraints*.

This implies that

$$\sum_{h=2}^{H+1} \sum_a q_\pi^h(s,a) = \sum_{h=2}^{H+1} \sum_{s',a} T(s \mid s', a) q_\pi^{h-1}(s', a)$$

$$\implies \sum_a q_\pi^1(s,a) + \sum_{h=2}^{H+1} \sum_a q_\pi^h(s,a) = \sum_{h=2}^{H+1} \sum_{s',a} T(s \mid s', a) q_\pi^{h-1}(s', a) + \sum_a q_\pi^1(s,a)$$

$$\implies \sum_a \sum_{h=1}^{H+1} q_\pi^h(s,a) = \sum_{h=2}^{H+1} \sum_{s',a} T(s \mid s', a) q_\pi^{h-1}(s', a) + \sum_a q_\pi^1(s,a)$$

$$\implies H \sum_a v_\pi^H(s,a) = \sum_{s',a} T(s \mid s', a) \left( \sum_{h=2}^{H+1} q_\pi^{h-1}(s', a) \right) + \mu(s)$$

$$\implies H \sum_a v_\pi^H(s,a) = H \sum_{s',a} T(s \mid s', a) v_\pi^H(s', a) + \mu(s)$$

$$\implies \sum_a v_\pi^H(s,a) = \sum_{s',a} T(s \mid s', a) v_\pi^H(s', a) + \frac{1}{H}\mu(s)$$

$$\implies u_\pi^H(s) \triangleq \sum_a v_\pi^H(s,a) = \sum_{s',a} T(s \mid s', a)\pi(a|s') u_\pi^H(s') + \frac{1}{H}\mu(s) \,. \tag{33}$$

Now, we denote $\mathbf{u}_\pi^H$ and $\bar{\mu}$ as corresponding column vectors and the transition matrix $\mathbf{P}^\pi \triangleq \left[ \sum_{s',a} T(s \mid s', a)\pi(a|s') \right]$. Thus, we obtain

$$(\mathbb{I} - \mathbf{P}^\pi) \mathbf{u}_\pi^H = \frac{1}{H}\bar{\mu}. \tag{34}$$

We can therefore express the finite horizon occupancy measure in matrix form as

$$\mathbf{v}_\pi^H = \mathbf{\Pi} \odot \left( (\mathbf{u}_\pi^H)^T \otimes \mathbf{1} \right)^T \tag{35}$$

where $\mathbf{\Pi}, \mathbf{v}_\pi \in \mathbb{R}^{|\mathcal{S}| \times |\mathcal{A}|}$, $\mathbf{1} \in \mathbb{R}^{|\mathcal{A}|}$ is a column vector of ones, $\otimes$ presents the Kronecker product, $\odot$ denotes the Hadamard product.

If we consider the case of a parameterized policy $\mathbf{\Pi}(\theta)$, the derivative of $\mathbf{v}_\pi^H$ with respect to $\theta$ is

$$
\begin{aligned}
\nabla_\theta \mathbf{v}_\pi^H &= \nabla_\theta \left[ \mathbf{\Pi} \odot \left( (\mathbf{u}_\pi^H)^T \otimes \mathbf{1} \right)^T \right] \\
&= \nabla_\theta \mathbf{\Pi} \odot \left( (\mathbf{u}_\pi^H)^T \otimes \mathbf{1} \right)^T + \mathbf{\Pi} \odot \nabla_\theta \left( (\mathbf{u}_\pi^H)^T \otimes \mathbf{1} \right)^T \\
&= \nabla_\theta \mathbf{\Pi} \odot \left( (\mathbf{u}_\pi^H)^T \otimes \mathbf{1} \right)^T + \mathbf{\Pi} \odot \left( (\nabla_\theta \mathbf{u}_\pi^H)^T \otimes \mathbf{1} \right)^T
\end{aligned}
\tag{36}
$$

The first term in Equation 36 is differentiable since the policy is parameterized by $\theta$. We show that $\nabla_\theta \mathbf{u}_\pi^H$ exists using Equation 34 as follows:

$$
\begin{aligned}
\nabla_\theta \left[ (\mathbb{I} - \mathbf{P}^\pi) \mathbf{u}_\pi^H \right] &= \frac{1}{H} \nabla_\theta \bar{\mu} \\
\nabla_\theta (\mathbb{I} - \mathbf{P}^\pi) \mathbf{u}_\pi^H + (\mathbb{I} - \mathbf{P}^\pi) \nabla_\theta \mathbf{u}_\pi^H &= 0 \\
-\nabla_\theta \mathbf{P}^\pi \mathbf{u}_\pi^H + (\mathbb{I} - \mathbf{P}^\pi) \nabla_\theta \mathbf{u}_\pi^H &= 0 \\
(\mathbb{I} - \mathbf{P}^\pi) \nabla_\theta \mathbf{u}_\pi^H &= \nabla_\theta \mathbf{P}^\pi \mathbf{u}_\pi^H
\end{aligned}
\tag{37}
$$

In Equation 37, we observe that $(\mathbb{I} - \mathbf{P}^\pi)^{-1}$ may or may not exist. To address this, we use the group inverse, a generalized matrix inverse that extends the concept of inversion to both singular and invertible matrices (Drazin, 1958). For a square matrix $\mathbf{C} \in \mathbb{R}^{n \times n}$, its group inverse $\mathbf{C}^\#$ satisfies the conditions $\mathbf{C}\mathbf{C}^\#\mathbf{C} = \mathbf{C}$, $\mathbf{C}^\#\mathbf{C}\mathbf{C}^\# = \mathbf{C}^\#$ and $\mathbf{C}\mathbf{C}^\# = \mathbf{C}^\#\mathbf{C}$.

The group inverse is useful in analysing Markov chains (Meyer, 1975) and coincides with the standard inverse when $\mathbf{C}$ is invertible. Using group inverse $(\mathbb{I} - \mathbf{P}^\pi)^\#$, Equation 37 can be expressed as,

$$
\nabla_\theta \mathbf{u}_\pi^H = (\mathbb{I} - \mathbf{P}^\pi)^\# \nabla_\theta \mathbf{P}^\pi \mathbf{u}_\pi^H + \mathbf{w}
\tag{38}
$$

where $\mathbf{w}$ is a vector in the null space of $(\mathbb{I} - \mathbf{P}^\pi)$. Note that $\mathbf{w} = \mathbf{0}$ for ergodic Markov chains (Meyer, 1975). An episodic MDP induces an ergodic Markov chain that admits a unique stationary distribution (Bojun, 2020) (see also Appendix B.3). Thus, Equation 38 simplifies to

$$
\nabla_\theta \mathbf{u}_\pi^H = (\mathbb{I} - \mathbf{P}^\pi)^\# \nabla_\theta \mathbf{P}^\pi \mathbf{u}_\pi^H
\tag{39}
$$

$$
= (\mathbb{I} - \mathbf{P}^\pi)^\# \left[ \sum_{s', a} T(s|s', a) \nabla_\theta \pi(a|s') \right] \mathbf{u}_\pi^H
\tag{40}
$$

which shows that $\nabla_\theta \mathbf{u}_\pi^H$ is differentiable and so is $\nabla_\theta \mathbf{v}_\pi^H$. Proceeding similarly, given the same conditions, we see that all higher derivatives of $\mathbf{v}_\pi^H$ also exist with respect to $\theta$. Thus, the space of parametrized finite-horizon occupancy measures $v_\pi^H$ forms a differentiable manifold $\mathcal{M}^H$.

### A.6 Optimal Transport Dataset Distance (OTDD)

Suppose we have two datasets, each consisting of feature-label pairs, $\mathcal{D}_A = \{(t_A^i, u_A^i)\}_{i=1}^m \sim P_A(t, u)$ and $\mathcal{D}_B = \{(t_B^i, u_B^i)\}_{i=1}^n \sim P_B(t, u)$ with $t_A, t_B \in \mathcal{T}$ and $u_A, u_B \in \mathcal{U}_A, \mathcal{U}_B$. These datasets can be used to create empirical distributions $\hat{P}_A(t, u)$ and $\hat{P}_B(t, u)$. OTDD is the p-Wasserstein distance between the datasets $\mathcal{D}_A$ and $\mathcal{D}_B$ - which is essentially the distance between their empirical distributions $\hat{P}_A$ and $\hat{P}_B$ - with the cost function defined as the metric of the joint space $\mathcal{T} \times \mathcal{U}$ (Alvarez-Melis & Fusi, 2020).

Naturally, the metric on this joint space can be defined as $d_{\mathcal{TU}}((t, u), (t', u')) = (d_\mathcal{T}(t, t')^p + d_\mathcal{U}(u, u')^p)^{1/p}$, for $p \geq 1$. However, in most applications $d_\mathcal{T}$ is readily available, while $d_\mathcal{U}$ might be scarce, especially in supervised learning (SL) between labels from unrelated label sets (Alvarez-Melis & Fusi, 2020). Further, we want $d_\mathcal{T}$ and $d_\mathcal{U}$ to have the same units to be addable. To overcome these issues, $d_\mathcal{U}$ is expressed in terms of $d_\mathcal{T}$ by mapping labels $u$ to distributions over the feature space $\mathcal{P}(\mathcal{T})$ as $u \to \alpha_u(T) \triangleq P(T \mid U = u) \in \mathcal{P}(\mathcal{T})$. Therefore, the distance between the labels $u$ and $u'$ is defined as the p-Wasserstein distance between $\alpha_u(T)$

and $\alpha_{u'}(T)$,

$$
\begin{aligned}
d_{\mathcal{U}}(u, u') &= \mathcal{W}_p(\alpha_u(T), \alpha_{u'}(T)) \\
&= \left( \min_{\pi \in \Pi(\alpha_u, \alpha_{u'})} \int_{\mathcal{T} \times \mathcal{T}} (d_{\mathcal{T}}(t, t'))^p \, d\pi(t, t') \right)^{1/p}
\end{aligned}
\tag{41}
$$

The metric on the joint space becomes,

$$
d_{\mathcal{TU}}((t, u), (t', u')) = \left( d_{\mathcal{T}}(t, t')^p + \mathcal{W}_p^p(\alpha_u(T), \alpha_{u'}(T)) \right)^{1/p}
\tag{42}
$$

Let $\mathcal{Z} = \mathcal{T} \times \mathcal{U}$, then the p-Wasserstein distance between $\hat{P}_A(t, u)$ and $\hat{P}_B(t, u)$ is a "nested" Wasserstein distance:

$$
\begin{aligned}
\mathcal{W}_p^p(\hat{P}_A, \hat{P}_B) &= \min_{\pi \in \Pi(P_A, P_B)} \int_{\mathcal{Z} \times \mathcal{Z}} (d_{\mathcal{Z}}(z, z'))^p \, d\pi \\
&= \min_{\pi \in \Pi(P_A, P_B)} \int_{\mathcal{TU} \times \mathcal{TU}} \left( d_{\mathcal{T}}(t, t')^p + \mathcal{W}_p^p(\alpha_u, \alpha_{u'}) \right) \, d\pi
\end{aligned}
\tag{43}
$$

$W_p^p(\hat{P}_A, \hat{P}_B)$ is the OTDD between datasets $\mathcal{D}_A$ and $\mathcal{D}_B$, often expressed as $d_{OT}(\mathcal{D}_A, \mathcal{D}_B)$. This is used in transfer learning to determine the distance (or similarity) between datasets. Figure 6 illustrates OTDD when applied to RL using datasets of state-action pairs.

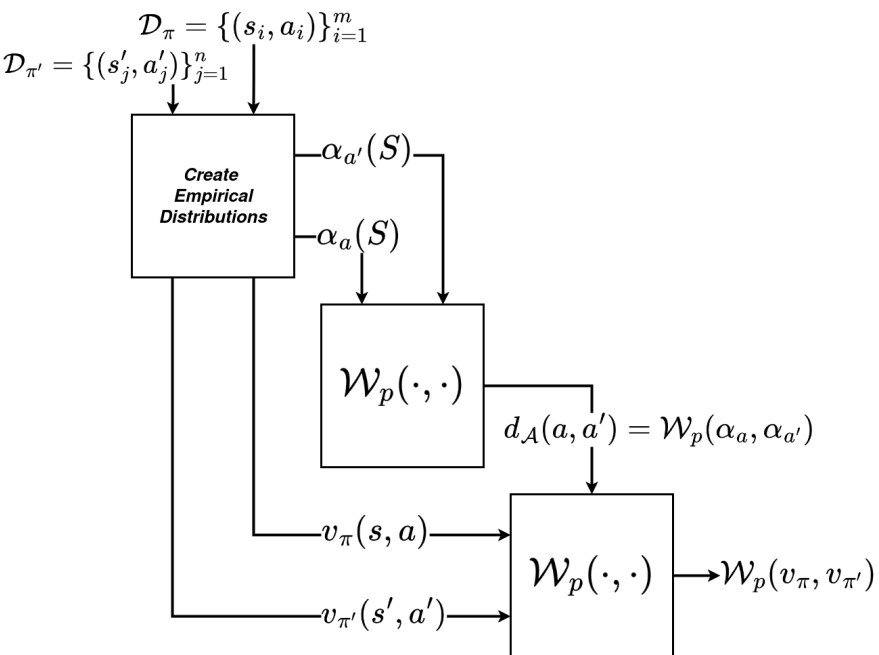

Figure 6: Illustration of OTDD when applied to RL.

### A.7 Proof of Proposition 4

We compute the error in occupancy measure for both the infinite and finite horizon cases. In infinite horizon MDPs, the occupancy measure is defined as the expected discounted number of visits of a state-action pair $(s, a)$ in a trajectory (Laroche & des Combes, 2023): $\mu = (1 - \gamma) \sum_{t=0}^{\infty} \gamma^t \mu_t$, where $\mu_t = P(s_t, a_t \mid \pi, \eta)$ is the state-action probability distribution at time step $t$ with the initial state distribution $\eta$ following the policy $\pi$. In finite horizon MDPs, the occupancy measure is the expected number of visits of a state-action pair $(s, a)$ in an episode of length $H$ (Altman, 1999): $\mu = \frac{1}{H} \sum_{t=1}^{H} \mu_t$.

First, we derive error bounds for the infinite horizon MDP in which $\gamma < 1$ and the occupancy measure is approximated using a finite number of samples collected up to a finite number of time steps $T$. Later, we derive error bounds for the finite horizon MDP.

### A.7.1 Infinite Horizon MDPs

**Estimated Occupancy Measure.** For convenience, we express the occupancy measure as $\mu = (1 - \gamma)\sum_{t=0}^{\infty}\gamma^t\mu_t$, where $\mu_t = P(s_t, a_t \mid \pi, \eta)$ is the state-action probability distribution at time step $t$ with the initial state distribution $\eta$ following the policy $\pi$. To compute $\mu$, we roll out $N$ episodes (each of multiple time steps) using $\pi$, and take $N$ number of samples at $t$ to approximate $\mu_t$. Thus, the empirical occupancy measure $\hat{\mu}$ is given by $\hat{\mu} = \rho\sum_{t=0}^{T}\gamma^t\hat{\mu}_t^N$, where $\rho = \frac{1}{\sum_{t=0}^{T}\gamma^t}$. Note that the total number of samples in the policy dataset $\mathcal{D}_\pi$ is $|\mathcal{D}_\pi| = N(T+1)$.

**Occupancy Measure Estimation Error.** Consider two occupancy measures $\mu = (1-\gamma)\sum_{t=0}^{\infty}\gamma^t\mu_t$ and $\nu = (1-\gamma)\sum_{t=0}^{\infty}\gamma^t\nu_t$ (with estimates $\hat{\mu} = \rho\sum_{t=0}^{T}\gamma^t\hat{\mu}_t^{N_\mu}$ and $\hat{\nu} = \rho\sum_{t=0}^{T}\gamma^t\hat{\nu}_t^{N_\nu}$). For independent sets $\{\mu_t\}_{t\geq0}$ and $\{\nu_t\}_{t\geq0}$, the Wasserstein distance has the following additive property (Panaretos & Zemel, 2019),

$$\mathcal{W}_p(\sum_t \mu_t, \sum_t \nu_t) \leq \sum_t \mathcal{W}_p(\mu_t, \nu_t) \tag{44}$$

While for $a \in \mathbb{R}$ (Panaretos & Zemel, 2019),

$$\mathcal{W}_p(a\mu, a\nu) = |a|\mathcal{W}_p(\mu, \nu) \tag{45}$$

Therefore, for our scenario where $p = 1$, the Wasserstein distance between $\mu$ and $\nu$ is given by:

$$\begin{aligned}
\mathcal{W}_1(\mu, \nu) &= \mathcal{W}_1((1-\gamma)\sum_{t=0}^{\infty}\gamma^t\mu_t, (1-\gamma)\sum_{t=0}^{\infty}\gamma^t\nu_t) \\
&\leq (1-\gamma)\sum_{t=0}^{\infty}\gamma^t\mathcal{W}_1(\mu_t, \nu_t)
\end{aligned} \tag{46}$$

while for $\hat{\mu}$ and $\hat{\nu}$,

$$\mathcal{W}_1(\hat{\mu}, \hat{\nu}) \leq \rho\sum_{t=0}^{T}\gamma^t\mathcal{W}_1(\hat{\mu}_t^{N_\mu}, \hat{\nu}_t^{N_\nu}) \tag{47}$$

In the RL problems we consider, the state-action space $\mathcal{Z} = \mathcal{S} \times \mathcal{A}$ is commonly defined as the subset of the Euclidean space $\mathcal{Z} \in \mathbb{R}^B$, where usually $B \geq 2$. Theorems 1 and 3 in (Sommerfeld et al., 2019) establish the following error bounds between the true and empirical probability distributions,

$$\begin{aligned}
\mathbb{E}[\mathcal{W}_1(\hat{\mu}_t^{N_\mu}, \mu_t)] &\leq \mathcal{E}_2 N_\mu^{-\frac{1}{2}} \\
\mathbb{E}[\mathcal{W}_1(\hat{\nu}_t^{N_\nu}, \nu_t)] &\leq \mathcal{E}_2 N_\nu^{-\frac{1}{2}}
\end{aligned} \tag{48}$$

where

$$\mathcal{E}_2 \leq 4B^{1/2}diam(\mathcal{Z}) \cdot \begin{cases} 2 + (1/2)\log_2|\mathcal{Z}| & \text{if } B = 2 \\ |\mathcal{Z}|^{1/2-1/B}\left[2 + 1/(2^{B/2-1} - 1)\right] & \text{if } B > 2 \end{cases}$$

Note that $|\mathcal{Z}|$ and $diam(\mathcal{Z})$ denote the cardinality and diameter of $\mathcal{Z}$, respectively.

Suppose $a = \mathcal{W}_1(\hat{\mu}, \hat{\nu})$, $b = \mathcal{W}_1(\hat{\mu}, \mu)$, $c = \mathcal{W}_1(\hat{\nu}, \mu)$, $d = \mathcal{W}_1(\mu, \nu)$, and $e = \mathcal{W}_1(\hat{\nu}, \nu)$. Then by performing two reverse triangle inequalities,

$$\begin{aligned}
&|a - c| \leq b \quad \text{and} \quad |c - d| \leq e \\
&\implies |a - d| \leq b + e
\end{aligned} \tag{49}$$

Equation 49 implies that,

$$
\begin{aligned}
\mathbb{E}[|\mathcal{W}_1(\hat{\mu}, \hat{\nu}) - \mathcal{W}_1(\mu, \nu)|] &\leq \mathbb{E}[\mathcal{W}_1(\hat{\mu}, \mu) + \mathcal{W}_1(\hat{\nu}, \nu)] \\
&= \mathbb{E}[\mathcal{W}_1(\rho \sum_{t=0}^{T} \gamma^t \hat{\mu}_t^{N_\mu}, \mu) + \mathcal{W}_1(\rho \sum_{t=0}^{T} \gamma^t \hat{\nu}_t^{N_\nu}, \nu)] \\
&= \mathbb{E}[\mathcal{W}_1(\rho \sum_{t=0}^{T} \gamma^t \hat{\mu}_t^{N_\mu}, \mu)] + \mathbb{E}[\mathcal{W}_1(\rho \sum_{t=0}^{T} \gamma^t \hat{\nu}_t^{N_\nu}, \nu)] \\
&\quad + \mathbb{E}[\mathcal{W}_1((1-\gamma) \sum_{t=0}^{\infty} \gamma^t \hat{\mu}_t^{N_\mu}, \mu) - \mathcal{W}_1((1-\gamma) \sum_{t=0}^{\infty} \gamma^t \hat{\mu}_t^{N_\mu}, \mu)] \\
&\quad + \mathbb{E}[\mathcal{W}_1((1-\gamma) \sum_{t=0}^{\infty} \gamma^t \hat{\nu}_t^{N_\nu}, \nu) - \mathcal{W}_1((1-\gamma) \sum_{t=0}^{\infty} \gamma^t \hat{\nu}_t^{N_\nu}, \nu)]
\end{aligned}
\tag{50}
$$

By virtue of triangle inequalities, we get

$$
\begin{aligned}
\mathcal{W}_1(\rho \sum_{t=0}^{T} \gamma^t \hat{\mu}_t^{N_\mu}, (1-\gamma) \sum_{t=0}^{\infty} \gamma^t \hat{\mu}_t^{N_\mu}) &\geq \mathcal{W}_1(\rho \sum_{t=0}^{T} \gamma^t \hat{\mu}_t^{N_\mu}, \mu) - \mathcal{W}_1((1-\gamma) \sum_{t=0}^{\infty} \gamma^t \hat{\mu}_t^{N_\mu}, \mu) \\
\mathcal{W}_1(\rho \sum_{t=0}^{T} \gamma^t \hat{\nu}_t^{N_\nu}, (1-\gamma) \sum_{t=0}^{\infty} \gamma^t \hat{\nu}_t^{N_\nu}) &\geq \mathcal{W}_1(\rho \sum_{t=0}^{T} \gamma^t \hat{\nu}_t^{N_\nu}, \nu) - \mathcal{W}_1((1-\gamma) \sum_{t=0}^{\infty} \gamma^t \hat{\nu}_t^{N_\nu}, \nu)
\end{aligned}
\tag{51}
$$

Therefore, the right-hand-side (R.H.S) of Equation 50 can be further simplified as

$$
\begin{aligned}
\text{R.H.S} &\leq \mathbb{E}[\mathcal{W}_1(\rho \sum_{t=0}^{T} \gamma^t \hat{\mu}_t^{N_\mu}, (1-\gamma) \sum_{t=0}^{\infty} \gamma^t \hat{\mu}_t^{N_\mu})] + \mathbb{E}[\mathcal{W}_1(\rho \sum_{t=0}^{T} \gamma^t \hat{\nu}_t^{N_\nu}, (1-\gamma) \sum_{t=0}^{\infty} \gamma^t \hat{\nu}_t^{N_\nu})] \\
&\quad + \mathbb{E}[\mathcal{W}_1((1-\gamma) \sum_{t=0}^{\infty} \gamma^t \hat{\mu}_t^{N_\mu}, \mu)] + \mathbb{E}[\mathcal{W}_1((1-\gamma) \sum_{t=0}^{\infty} \gamma^t \hat{\nu}_t^{N_\nu}, \nu)]
\end{aligned}
\tag{52}
$$

For simplicity, we denote $\hat{\mu}_\infty = (1-\gamma) \sum_{t=0}^{\infty} \gamma^t \hat{\mu}_t^{N_\mu}$ (similarly $\hat{\nu}_\infty$) and $\hat{\mu}_T = \rho \sum_{t=0}^{T} \gamma^t \hat{\mu}_t^{N_\mu}$ (similarly $\hat{\nu}_T$), where $\rho = \frac{1}{\sum_{t=0}^{T} \gamma^t} = \frac{1-\gamma}{1-\gamma^{T+1}}$. Using Theorem 4 in (Gibbs & Su, 2002), the 1-Wasserstein metric $\mathcal{W}_1$ and the total variation distance $d_{TV}$ satisfy the following,

$$
\begin{aligned}
\mathcal{W}_1(\hat{\mu}_\infty, \hat{\mu}_T) &\leq diam(\mathcal{Z}) \cdot d_{TV}(\hat{\mu}_\infty, \hat{\mu}_T) \\
&= diam(\mathcal{Z}) \cdot \frac{1}{2} \sum_{z \in \mathcal{Z}} |\hat{\mu}_\infty(z) - \hat{\mu}_T(z)|
\end{aligned}
\tag{53}
$$

However,

$$\hat{\mu}_\infty - \hat{\mu}_T = (1-\gamma)\sum_{t=0}^{\infty}\gamma^t\hat{\mu}_t^{N_\mu} - \frac{1-\gamma}{1-\gamma^{T+1}}\sum_{t=0}^{T}\gamma^t\hat{\mu}_t^{N_\mu}$$

$$= (1-\gamma)\sum_{t=0}^{\infty}\gamma^t\hat{\mu}_t^{N_\mu} - \frac{1-\gamma}{1-\gamma^{T+1}}\sum_{t=0}^{T}\gamma^t\hat{\mu}_t^{N_\mu}$$

$$+ (1-\gamma)\sum_{t=0}^{T}\gamma^t\hat{\mu}_t^{N_\mu} - (1-\gamma)\sum_{t=0}^{T}\gamma^t\hat{\mu}_t^{N_\mu}$$

$$= (1-\gamma)\left(\sum_{t=0}^{\infty}\gamma^t\hat{\mu}_t^{N_\mu} - \sum_{t=0}^{T}\gamma^t\hat{\mu}_t^{N_\mu}\right) + \left((1-\gamma) - \frac{1-\gamma}{1-\gamma^{T+1}}\right)\sum_{t=0}^{T}\gamma^t\hat{\mu}_t^{N_\mu} \qquad (54)$$

$$= (1-\gamma)\sum_{t=T+1}^{\infty}\gamma^t\hat{\mu}_t^{N_\mu} - \gamma^{T+1}\frac{1-\gamma}{1-\gamma^{T+1}}\sum_{t=0}^{T}\gamma^t\hat{\mu}_t^{N_\mu}$$

$$\leq (1-\gamma)\sum_{t=T+1}^{\infty}\gamma^t\hat{\mu}_t^{N_\mu}$$

$$= \gamma^{T+1}\frac{1-\gamma}{\gamma^{T+1}}\sum_{t=T+1}^{\infty}\gamma^t\hat{\mu}_t^{N_\mu}$$

$$= \gamma^{T+1}\hat{\mu}_{T+1,\infty}$$

where $\frac{1-\gamma}{\gamma^{T+1}}$ normalizes $\sum_{t=T+1}^{\infty}\gamma^t\hat{\mu}_t^{N_\mu}$. We utilize Equation 54 in Equation 53 as,

$$\mathcal{W}_1(\hat{\mu}_\infty, \hat{\mu}_T) \leq diam(\mathcal{Z})\cdot\frac{1}{2}\sum_{z\in\mathcal{Z}}|\hat{\mu}_\infty(z) - \hat{\mu}_T(z)|$$

$$\leq diam(\mathcal{Z})\cdot\frac{1}{2}\sum_{z\in\mathcal{Z}}|\gamma^{T+1}\hat{\mu}_{T+1,\infty}(z)| \qquad (55)$$

$$= \frac{\gamma^{T+1}}{2}diam(\mathcal{Z})$$

Equation 55 also applies for $\mathcal{W}_1(\hat{\nu}_\infty, \hat{\nu}_T)$, therefore by substituting these into Equation 52,

$$\text{R.H.S} \leq \mathbb{E}[\mathcal{W}_1((1-\gamma)\sum_{t=0}^{\infty}\gamma^t\hat{\mu}_t^{N_\mu}, \mu)] + \mathbb{E}[\mathcal{W}_1((1-\gamma)\sum_{t=0}^{\infty}\gamma^t\hat{\nu}_t^{N_\nu}, \nu)] + \gamma^{T+1}diam(\mathcal{Z})$$

$$= \mathbb{E}[\mathcal{W}_1((1-\gamma)\sum_{t=0}^{\infty}\gamma^t\hat{\mu}_t^{N_\mu}, (1-\gamma)\sum_{t=0}^{\infty}\gamma^t\mu_t)]$$

$$+ \mathbb{E}[\mathcal{W}_1((1-\gamma)\sum_{t=0}^{\infty}\gamma^t\hat{\nu}_t^{N_\nu}, (1-\gamma)\sum_{t=0}^{\infty}\gamma^t\nu_t)] + \gamma^{T+1}diam(\mathcal{Z}) \qquad (56)$$

$$\leq (1-\gamma)\sum_{t=0}^{\infty}\gamma^t\left(\mathbb{E}[\mathcal{W}_1(\hat{\mu}_t^{N_\mu}, \mu_t)] + \mathbb{E}[\mathcal{W}_1(\hat{\nu}_t^{N_\mu}, \nu_t)]\right) + \gamma^{T+1}diam(\mathcal{Z}).$$

By substituting Equation 48 into Equation 56

$$\text{R.H.S} \leq (1-\gamma)\sum_{t=0}^{\infty}\gamma^t\left(\mathcal{E}_2 N_\mu^{-\frac{1}{2}} + \mathcal{E}_2 N_\nu^{-\frac{1}{2}}\right) + \gamma^{T+1}diam(\mathcal{Z})$$

$$= \mathcal{E}_2\left(N_\mu^{-\frac{1}{2}} + N_\nu^{-\frac{1}{2}}\right) + \gamma^{T+1}diam(\mathcal{Z}) \qquad (57)$$

Therefore, Equation 50 becomes:

$$\mathbb{E}[|\mathcal{W}_1(\hat{\mu}, \hat{\nu}) - \mathcal{W}_1(\mu, \nu)|] \leq \mathcal{E}_2 \left( N_\mu^{-\frac{1}{2}} + N_\nu^{-\frac{1}{2}} \right) + \gamma^{T+1} diam(\mathcal{Z}) \tag{58}$$

**Over the full trajectory in the occupancy measure space.** The true distance between consecutive policies $\pi_i$ and $\pi_{i+1}$ after an update is $\mathcal{W}_1(v_{\pi_i}, v_{\pi_{i+1}})$, which is induced by the $i^{th}$ policy update. We estimate this distance using datasets of the policies, i.e. approximated distributions, using $\mathcal{W}_1(\hat{v}_{\pi_i}, \hat{v}_{\pi_{i+1}})$.

For $M$ roll out episodes of each $\pi_i$, we use Equation 58, with $N_\mu = N_\nu = M$, to derive the following error bounds,

$$\mathbb{E}\left[ \left| \mathcal{W}_1(v_{\pi_i}, v_{\pi_{i+1}}) - \mathcal{W}_1(\hat{v}_{\pi_i}, \hat{v}_{\pi_{i+1}}) \right| \right] \leq 2\mathcal{E}_2 M^{-\frac{1}{2}} + \gamma^{T+1} diam(\mathcal{Z}) \tag{59}$$

which is consistent with learning from $\mathcal{D}_{\pi_i}$ and then $\mathcal{D}_{\pi_{i+1}}$. By summing sequentially through policies encountered during RL training, we compute the total distance over a path of $N$ segments obtained via policy updates:

$$\sum_{i=0}^{N-1} \mathbb{E}\left[ \left| \mathcal{W}_1(v_{\pi_i}, v_{\pi_{i+1}}) - \mathcal{W}_1(\hat{v}_{\pi_i}, \hat{v}_{\pi_{i+1}}) \right| \right] \leq 2N\mathcal{E}_2 M^{-\frac{1}{2}} + N\gamma^{T+1} diam(\mathcal{Z}) \tag{60}$$

Since $|\sum_t x_t| \leq \sum_t |x_t|$ then,

$$\mathbb{E}\left[ \left| \sum_{i=0}^{N-1} \mathcal{W}_1(v_{\pi_i}, v_{\pi_{i+1}}) - \sum_{i=0}^{N-1} \mathcal{W}_1(\hat{v}_{\pi_i}, \hat{v}_{\pi_{i+1}}) \right| \right] \leq \frac{2N\mathcal{E}_2}{\sqrt{M}} + N\gamma^{T+1} diam(\mathcal{Z}) \tag{61}$$

### A.7.2 Finite Horizon MDPs

**Occupancy Measure Estimated Error.** Consider two occupancy measures $\mu = \frac{1}{H}\sum_{t=1}^{H} \mu_t$ and $\nu = \frac{1}{H}\sum_{t=1}^{H} \nu_t$ with estimates $\hat{\mu} = \frac{1}{H}\sum_{t=1}^{H} \hat{\mu}_t^{N_\mu}$ and $\hat{\nu} = \frac{1}{H}\sum_{t=1}^{H} \hat{\nu}_t^{N_\nu}$. From Equation 49, we have

$$\begin{aligned}
&\mathbb{E}[|\mathcal{W}_1(\hat{\mu}, \hat{\nu}) - \mathcal{W}_1(\mu, \nu)|] \\
&\leq \mathbb{E}[\mathcal{W}_1(\hat{\mu}, \mu) + \mathcal{W}_1(\hat{\nu}, \nu)] \\
&= \mathbb{E}[\mathcal{W}_1(\frac{1}{H}\sum_{t=1}^{H}\hat{\mu}_t^{N_\mu}, \frac{1}{H}\sum_{t=1}^{H}\mu_t) + \mathcal{W}_1(\frac{1}{H}\sum_{t=1}^{H}\hat{\nu}_t^{N_\nu}, \frac{1}{H}\sum_{t=1}^{H}\nu_t)] \\
&\leq \frac{1}{H}\sum_{t=1}^{H}\mathbb{E}[\mathcal{W}_1(\hat{\mu}_t^{N_\mu}, \mu_t)] + \frac{1}{H}\sum_{t=1}^{H}\mathbb{E}[\mathcal{W}_1(\hat{\nu}_t^{N_\nu}, \nu_t)] \\
&\leq \mathcal{E}_2 \left( N_\mu^{-\frac{1}{2}} + N_\nu^{-\frac{1}{2}} \right)
\end{aligned} \tag{62}$$

Therefore **for the total path in the occupancy measure space** with $M$ roll out episodes of each $\pi_i$, the error bound is

$$\mathbb{E}\left[ \left| \sum_{i=0}^{N-1} \mathcal{W}_1(v_{\pi_i}, v_{\pi_{i+1}}) - \sum_{i=0}^{N-1} \mathcal{W}_1(\hat{v}_{\pi_i}, \hat{v}_{\pi_{i+1}}) \right| \right] \leq \frac{2N\mathcal{E}_2}{\sqrt{M}} \tag{63}$$

by assigning $N_\mu = N_\nu = M$ in Equation 62, which concludes the proof.

## A.8   Proof of Proposition 5

By definition of $\eta_{sub}$, we get

$$
\begin{aligned}
\eta_{sub} &= \frac{\sum_{i=0}^{N-2} \mathcal{W}_1(v_{\pi_i}, v_{\pi_{i+1}}) + \mathcal{W}_1(v_{\pi_{N-1}}, v_{\pi_N})}{\mathcal{W}_1(v_{\pi_0}, v_{\pi_N})} \\
&= \frac{\sum_{i=0}^{N-2} \mathcal{W}_1(v_{\pi_i}, v_{\pi_{i+1}}) + \mathcal{W}_1(v_{\pi_{N-1}}, v_{\pi_N})}{\mathcal{W}_1(v_{\pi_0}, v_{\pi^*})} \times \frac{\mathcal{W}_1(v_{\pi_0}, v_{\pi^*})}{\mathcal{W}_1(v_{\pi_0}, v_{\pi_N})} \\
&\geq \frac{\sum_{i=0}^{N-2} \mathcal{W}_1(v_{\pi_i}, v_{\pi_{i+1}}) + \mathcal{W}_1(v_{\pi_{N-1}}, v_{\pi^*}) - \mathcal{W}_1(v_{\pi_N}, v_{\pi^*})}{\mathcal{W}_1(v_{\pi_0}, v_{\pi^*})} \times \frac{\mathcal{W}_1(v_{\pi_0}, v_{\pi^*})}{\mathcal{W}_1(v_{\pi_0}, v_{\pi_N})} \\
&= \left( \eta - \frac{\mathcal{W}_1(v_{\pi_N}, v_{\pi^*})}{\mathcal{W}_1(v_{\pi_0}, v_{\pi_N})} \right) \frac{\mathcal{W}_1(v_{\pi_0}, v_{\pi^*})}{\mathcal{W}_1(v_{\pi_0}, v_{\pi_N})} .
\end{aligned}
\tag{64}
$$

The inequality above is true due to the triangle inequality $\mathcal{W}_1(v_{\pi_{N-1}}, v_{\pi_N}) + \mathcal{W}_1(v_{\pi_N}, v_{\pi^*}) \geq \mathcal{W}_1(v_{\pi_{N-1}}, v_{\pi^*})$. By applying triangle inequality, we also get

$$
\mathcal{W}_1(v_{\pi_0}, v_{\pi^*}) + \mathcal{W}_1(v_{\pi_N}, v_{\pi^*}) \geq \mathcal{W}_1(v_{\pi_0}, v_{\pi_N}) .
$$

This implies that

$$
\frac{\mathcal{W}_1(v_{\pi_0}, v_{\pi^*})}{\mathcal{W}_1(v_{\pi_0}, v_{\pi_N})} \geq 1 - \frac{\mathcal{W}_1(v_{\pi_N}, v_{\pi^*})}{\mathcal{W}_1(v_{\pi_0}, v_{\pi_N})} .
\tag{65}
$$

Equation 64 and Equation 65 together yield

$$
\begin{aligned}
\eta_{sub} &\geq \left( \eta - \frac{\mathcal{W}_1(v_{\pi_N}, v_{\pi^*})}{\mathcal{W}_1(v_{\pi_0}, v_{\pi_N})} \right) \left( 1 - \frac{\mathcal{W}_1(v_{\pi_N}, v_{\pi^*})}{\mathcal{W}_1(v_{\pi_0}, v_{\pi_N})} \right) \\
&= \eta - \frac{\mathcal{W}_1(v_{\pi_N}, v_{\pi^*})}{\mathcal{W}_1(v_{\pi_0}, v_{\pi_N})} - \eta \frac{\mathcal{W}_1(v_{\pi_N}, v_{\pi^*})}{\mathcal{W}_1(v_{\pi_0}, v_{\pi_N})} + \left( \frac{\mathcal{W}_1(v_{\pi_N}, v_{\pi^*})}{\mathcal{W}_1(v_{\pi_0}, v_{\pi_N})} \right)^2 \\
&\geq \eta \left( 1 - \frac{\mathcal{W}_1(v_{\pi_N}, v_{\pi^*})}{\mathcal{W}_1(v_{\pi_0}, v_{\pi_N})} \right) - \frac{\mathcal{W}_1(v_{\pi_N}, v_{\pi^*})}{\mathcal{W}_1(v_{\pi_0}, v_{\pi_N})} \\
&\geq \eta \left( 1 - \frac{2\mathcal{W}_1(v_{\pi_N}, v_{\pi^*})}{\mathcal{W}_1(v_{\pi_0}, v_{\pi_N})} \right) .
\end{aligned}
$$

The second last inequality is due to non-negativity of $\left( \frac{\mathcal{W}_1(v_{\pi_N}, v_{\pi^*})}{\mathcal{W}_1(v_{\pi_0}, v_{\pi_N})} \right)^2$. The last inequality is due to the fact that $\eta \geq 1$.

Thus, we conclude that

$$
\frac{\eta - \eta_{sub}}{\eta} \leq \frac{2\mathcal{W}_1(v_{\pi_N}, v_{\pi^*})}{\mathcal{W}_1(v_{\pi_0}, v_{\pi_N})} .
$$

## A.9   Wasserstein Spaces as Geodesic Spaces

Given probability measures $\mu, \nu \in \mathcal{P}(\mathcal{X})$ on a metric space $\mathcal{X} \subset \mathbb{R}^B$ with metric $d_{\mathcal{X}}(x, x')$, the Wasserstein distance $\mathcal{W}_p(\mu, \nu)$ is the minimal transport cost for $c(x, x') = (d_{\mathcal{X}}(x, x'))^p$ with $p \geq 1$ (Villani, 2009). The Wasserstein distance $\mathcal{W}_p(\mu, \nu)$ takes a distance on $\mathcal{X}$ and creates out of it a distance on $\mathcal{P}(\mathcal{X})$(Peyré, 2019). Proposition 5.1 in (Santambrogio, 2015) asserts that $\mathcal{W}_p$ is a distance over $\mathcal{P}(\mathcal{X})$.

**Definition A.9** (Wasserstein Space). (Santambrogio, 2015) *Given a Polish space $\mathcal{X}$, for each $p \in [1, \infty)$, the space $\mathcal{P}(\mathcal{X})$ endowed with the distance $\mathcal{W}_p$ is a Wasserstein space $\mathbb{W}_p$ of order $p$.*

Theorem 5.27 in (Santambrogio, 2015) states that if $\mathcal{X}$ is a convex space, then the space $\mathbb{W}_p$ is a geodesic space (length space). Thus, the geodesic (shortest path distance) between $\mu, \nu \in \mathcal{P}(\mathcal{X})$ is given by $\mathcal{W}_p(\mu, \nu)$

(Kolouri et al., 2017). It was mentioned in Appendix A.7.1 that the RL problems we consider consist of the state-action space $\mathcal{Z} = \mathcal{S} \times \mathcal{A} \in \mathbb{R}^B : B \geq 2$ (subsets of the Euclidean space). Given that Euclidean spaces are convex spaces (Boyd & Vandenberghe, 2004), our space of occupancy measures $\mathcal{M}$ is a Wasserstein space $\mathbb{W}_1 = (\mathcal{M}, \mathcal{W}_1)$ and thus a geodesic space. Therefore, $\mathcal{W}_1(\mu, \nu)$ measures the shortest path on the surface of the manifold $\mathcal{M}$ between probability distributions $\mu$ and $\nu$.

# B    Additional Experimental Analysis and Results

## B.1    Environment Description

**2D-Gridworld** environment of size 5x5 with actions: {up, right, down, left}. The start and goal states are always located at top-left and bottom-right, respectively. In the gridworld, we perform experiments on three settings namely:- A) *Deterministic, dense rewards setting*: State transitions are deterministic. The reward function is given by $\|s_t - s_g\|_1$, where $s_t$ is the agent state at timestep $t$ and $s_g$ is the goal state. B) *Deterministic, sparse rewards setting*: State transitions are deterministic and all states issue a reward of -0.04 except the goal state with reward of 1. C) *Stochastic, dense rewards setting*: Actions have a probability of 0.8 in the instructed direction and 0.1 in each adjacent direction. Reward function is as defined in setting A.

**2D-Gridworlds (Task Difficulty).** Figure 7 depicts the configurations of the 5 tasks that were used to assess ESL with respect to task hardness. They are all deterministic with actions: {up, right, down, left}, and mostly have the start-state at the top-left and the goal-state at the bottom-right - with only one task that has the goal-state at the center. In the order of appearance: a) *[5x5] dense*: has size 5x5 and dense rewards, b) *[5x5] sparse (hard)*: has size 5x5 and sparse rewards, c) *[5x5] sparse (easy)*: has size 5x5, sparse rewards, and goal-state at the center, d) *[15x15] dense*: has size 15x15 and dense rewards, and e) *[15x15] sparse*: has size 15x15 and sparse rewards. The reward functions for both dense and sparse rewards are as previously described above for **2D-Gridworld**.

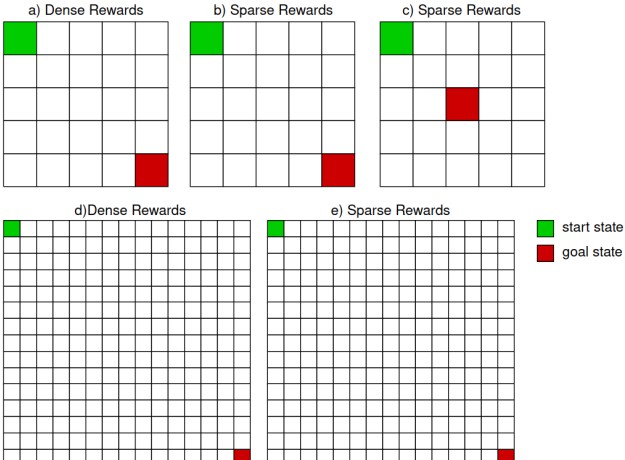

Figure 7: Five gridworld tasks with the same action space, but different rewards, state space and location of the goal state.

## B.2    OMR(k): OMR over number of updates

OMR is defined for the entire policy trajectory by Equation 6 as,

$$\kappa \triangleq \frac{\sum_{k \in K^+} \mathcal{W}_1(v_{\pi_k}, v_{\pi_{k+1}})}{\sum_{k=0}^{N-1} \mathcal{W}_1(v_{\pi_k}, v_{\pi_{k+1}})} \ .$$

To observe how it changes with respect to updates, we compute OMR from update $i$ onwards till the end of the learning trajectory, i.e. over subsequences with a decreasing number of policy updates with increasing $i$, using:

$$\kappa(i) \triangleq \frac{\sum_{k \in K^+, k \geq i} \mathcal{W}_1(v_{\pi_k}, v_{\pi_{k+1}})}{\sum_{k=i}^{N-1} \mathcal{W}_1(v_{\pi_k}, v_{\pi_{k+1}})}, \text{ such that } i \in [0, N - T] \tag{66}$$

where $T \approx 0.9N$ to ensure that the last subsequence of policy updates have at least 10% of the total updates in the trajectory.

## B.3 Computation of Occupancy Measures

The finite-horizon occupancy measure is defined as (Altman, 1999),

$$v_\pi^H(s, a) = \frac{1}{H} \sum_{t=1}^{H} \mathbb{P}(s_t = s, a_t = a \mid \pi, \mu)$$

for which the probability of the state-action pair selected is time-dependent. If we restrict our analysis to stationary policies where $\pi(a_t|s_t) = \pi(a|s)$, then the probability of the state-action pair becomes time-independent and thus,

$$v_\pi^H(s, a) = \mathbb{P}(s, a \mid \pi, \mu)$$

This implies that the use of stationary policies in finite-horizon MDPs, as observed in practice with many episodic MDPs (Memmel et al., 2022; Aleksandrowicz & Jaworek-Korjakowska, 2023; Liu, 2023), induces stationary occupancy measures - where the expected number of state-action pair visits are independent of the time-step. Work by (Bojun, 2020) provides extensive details about the existence of stationarity in episodic MDPs and shows (in Theorem 4) that,

$$\mathbb{E}_{(s,a) \sim v_\pi^H} \left[ \bar{\mathcal{R}}(s, a) \right] = \frac{\mathbb{E}_{\zeta \sim M_\pi} \left[ \sum_{t=1}^{H(\zeta)} R(s_t, a_t) \right]}{\mathbb{E}_{\zeta \sim M_\pi} \left[ H(\zeta) \right]} \tag{67}$$

where $\zeta$ is the episodic state-action pair trajectory, $H(\zeta)$ is the episode length corresponding to $\zeta$, and $M_\pi$ is the markov chain induced by policy $\pi$. We verified the correctness of our $v_\pi^H$ computation by calculating the relative error derived from Equation 67 to check its validity. The relative error is given as

$$\text{Rel. Error \%} = 100 * \frac{\mathbb{E}_{(s,a) \sim v_\pi^H} \left[ \bar{\mathcal{R}}(s, a) \right] \mathbb{E}_{\zeta \sim M_\pi} \left[ H(\zeta) \right] - \mathbb{E}_{\zeta \sim M_\pi} \left[ \sum_{t=1}^{H(\zeta)} R(s_t, a_t) \right]}{\mathbb{E}_{\zeta \sim M_\pi} \left[ \sum_{t=1}^{H(\zeta)} R(s_t, a_t) \right]} \tag{68}$$

Figure 8 depicts Rel. Error% vs number of updates in the stochastic 2D-Gridworld environment with dense rewards. We observe that increasing the number of rollouts $M$ reduces the estimation error of $v_\pi^H$. For $M = 10$, the absolute relative error can be as high as 50% with the mean less than 10%. While for $M = 500$, the maximum absolute relative error is 4%.

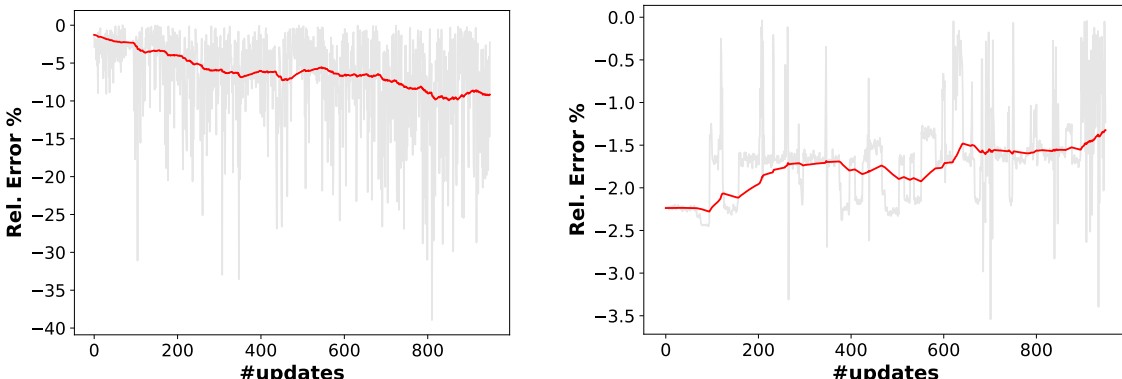

Figure 8: Rel. Error% vs number of updates plots in the 2D-Gridworld environment where $v_\pi^H$ is estimated using $M = 10$ rollouts (left) and $M = 500$ rollouts (right).

### B.4 Effects of the number of rollouts - SAC

The policy dataset $D_{\pi_i}$ in a deterministic environment is made up of (s,a) pairs generated from a single episode of the policy $\pi_i$. In a deterministic environment, this sequence remains the same across repeats of episodes, for each policy $\pi_i$ (deterministic) at update step $i$. Therefore, a single rollout is sufficient to estimate the occupancy measure $v_{\pi_i}$. In a stochastic environment, rollouts are impacted by the environment's stochasticity. Thus, multiple rollouts are needed to estimate the occupancy measure accurately. As the number of rollouts increases, the occupancy measure should converge and become less noisy.

Table 5 shows that, in a stochastic setting, the ESL values converge as the number of rollouts increases. OMR appears to be invariant across various the number of rollouts, and the mean number of updates appear to be consistent around 2900 (with exception for #rollouts = 1). The results indicate that from about 6 rollouts, the estimated occupancy measures become less noisy. This aligns with Equation 61, which shows that increasing the number of rollouts reduces estimation error.

| #rollouts | ESL | OMR | UC |
|---|---|---|---|
| 1 | 849.1±468.5 | 0.500±0.004 | 1849±742.2 |
| 3 | 618.6±257.3 | 0.501±0.005 | 2413±1397 |
| 6 | 445.4±245.8 | 0.501±0.042 | 2462±2043 |
| 9 | 428.1±234.4 | 0.503±0.004 | 2281±1743 |

Table 5: Evaluation of SAC algorithm in the **stochastic, dense-rewards setting** for 5x5 gridworld with **40 maximum steps per episode** across various number of rollouts. The effects of #rollouts on the Effort of Sequential Learning (ESL), Optimal Movement Ratio (OMR), and number of updates to convergence (UC) are observed.

### B.5 $\eta_{sub}$ can be a reasonable proxy for $\eta$, when optimal policy is not fully reached

We compare ESL when the optimal policy was reached, denoted $\eta$, versus when it was not, denoted $\eta_{sub}$, in Tables 6 and 7. First, we observe that the number of rollouts impacts the metric values. Second, $\eta_{sub}$ values are always greater than $\eta$ values. Note that UCRL2 and PSRL update their policies only at the end of each episode, whereas SAC and DQN update theirs after each time step. Hence, $UC_{sub} = 499$ for both UCRL2 and PSRL.

The ESL values (both $\eta$ and $\eta_{sub}$) in Table 7 are lower than those in Table 6, as expected since more data samples reduce estimation error. The distance from the initial policies to the final polices are not so different. Using both Tables 6 and 7, we notice that comparing algorithms with $\eta_{sub}$ yields the same efficiency ranking (e.g. PSRL, UCRL, SAC and DQN) as $\eta$. This indicates that $\eta_{sub}$ reliably predicts results provided by $\eta$ for comparing algorithms.

The results presented in Table 2 for stochastic dense-rewards setting are consistent with those in Table 7 because the number of rollouts used was Nr = 6.

### B.6 Effects of Hyperparameters - UCRL2

Table 8 illustrates the effects of hyperparameter values in the UCRL2 algorithm. The environment is deterministic dense-rewards setting with 200 training episodes. We observe that high exploration rates ($\delta \to 0$) appear to align with high ESL and UC, while high exploitation rates ($\delta \to 1$) appear to align with low ESL and UC. OMR appears to be invariant across various $\delta$ values.

| Algo. | $\eta$ | $\eta_{sub}$ | d | c | UC | $UC_{sub}$ |
|-------|--------|--------------|---|---|-----|-----------|
| SAC | 849± 468 | 3623± 4166 | 5.63± 1.50 | 5.26± 2.10 | 1850± 742 | 7451± 3535 |
| UCRL2 | 230± 155 | 613± 999 | 5.65± 0.93 | 5.45± 2.15 | 284± 180 | 499± 0.0 |
| PSRL | 86.2± 44.4 | 389± 102 | 4.96± 1.26 | 5.29± 1.49 | 97.2± 52.5 | 499± 0.0 |
| DQN | 564± 478 | 3911± 1710 | 5.52± 1.39 | 6.54± 2.05 | 1213± 1061 | 9097± 1904 |

Table 6: Evaluation of algorithms in the **stochastic, dense-rewards setting** for 5x5 gridworld with **40 maximum steps per episode** with the number of rollouts Nr = 1. The total number of training episodes is 500. When the algorithm converged at optimality, $\eta$ is the *Effort of Sequential Learning*, $d = \mathcal{W}_1(\pi_0, \pi^*)$ is distance from initial policy to the optimal policy, and UC is the number of updates to convergence. When the algorithm did not converge at the optimal policy, rather a non-optimal $\pi_N$, we use $\eta_{sub}$, $c = \mathcal{W}_1(\pi_0, \pi_N)$, and $UC_{sub}$ to denote the aforementioned quantities. 40 training trials were used.

| Algo. | $\eta$ | $\eta_{sub}$ | d | c | UC | $UC_{sub}$ |
|-------|--------|--------------|---|---|-----|-----------|
| SAC | 445± 246 | 853± 127 | 5.63± 1.23 | 7.26± 1.45 | 2463± 2043 | 6293± 441 |
| UCRL2 | 198± 121 | 510± 274 | 5.36± 0.84 | 4.58± 1.90 | 268± 155 | 499± 0.0 |
| PSRL | 55.4± 33.6 | 361± 43.6 | 4.97± 1.34 | 3.91± 0.48 | 76.1± 50.6 | 499± 0.0 |
| DQN | 458± 311 | 1971± 250 | 4.88± 1.06 | 6.52± 0.31 | 1586± 1077 | 13713± 6907 |

Table 7: Evaluation of algorithms in the **stochastic, dense-rewards setting** for 5x5 gridworld with **40 maximum steps per episode** with the number of rollouts Nr = 6. The total number of training episode is 500. When the algorithm converged at optimality, $\eta$ is the *Effort of Sequential Learning*, $d = \mathcal{W}_1(\pi_0, \pi^*)$ is distance from initial policy to the optimal policy, and UC is the number of updates to convergence. When the algorithm did not converge at the optimal policy however some $\pi_N$, we use $\eta_{sub}$, $c = \mathcal{W}_1(\pi_0, \pi_N)$, and $UC_{sub}$ to denote the aforementioned quantities. 40 training trials were used.

| $\delta$ | ESL | OMR | UC | SR% |
|----------|-----|-----|-----|-----|
| 0.1 | 47.76±7.768 | 0.512±0.033 | 62.26±9.977 | 100 |
| 0.3 | 39.29±5.860 | 0.515±0.034 | 58.08±7.746 | 100 |
| 0.5 | 38.26±6.747 | 0.511±0.036 | 56.92±9.111 | 100 |
| 0.7 | 37.48±5.094 | 0.507±0.029 | 56.68±7.460 | 100 |
| 0.9 | 36.40±5.301 | 0.510±0.036 | 54.86±7.326 | 100 |

Table 8: Evaluation of UCRL2 algorithm in the **deterministic, dense-rewards setting** for 5x5 gridworld with **15 maximum steps per episode**. Different confidence parameter $\delta \in (0, 1)$ were evaluated to see their effects on Effort of Sequential Learning (ESL), Optimal Movement Ratio (OMR), number of updates to convergence (UC), and success rate (SR). Note that as $\delta \to 0$, the agent approaches absolute exploration, and with $\delta \to 1$ absolute exploitation.

### B.7 Extended Discussion of Usefulness of ESL and OMR

The quantities like regret and number of updates (UC) are outcomes of the exploratory processes, and thus reflect only a partial view of the underlying exploration mechanisms. We propose ESL and OMR to complement regret and number of updates as metrics but not to replace them.

**1. Complementarity of ESL and OMR with respect to UC:**

a. **Case 1.** Let us consider two RL algorithms that reach optimality with the same number of updates, i.e. they have the same UC. *How would one be able to distinguish the exploratory processes of these algorithms?*

ESL and OMR are the summary metrics of the policy trajectory during learning. These can reveal which algorithm's exploratory process is more direct versus meandering, smooth versus noisy, or has large versus small coverage area in the policy space (Figures 3 and 4, top rows). Therefore, ESL and OMR quantify with granularity the characteristics of the exploratory process of an RL algorithm for any given environment.

b. **Case 2.** Let us consider the case when optimality is not reached but the maximum number of updates is attained by two RL algorithms. *How would one be able to evaluate the exploratory processes of these algorithms and systematically uncover which exploratory process demonstrates desired characteristics?* Looking into the training trajectories of RL algorithms in an environment and corresponding higher/lower ESLs ($\eta_{sub}$, Section 4.2), we can make a knowledgeable choice of an RL algorithm exhibiting desired characteristics (e.g. high coverage, smooth exploration). We have shown in Section 4.2 and results in Section 5.3 (also Appendix B.5) that ranking based on suboptimal ESL is aligned with true ESL, and additionally, the visualization of the training trajectories (Figures 3 and 4) can indicate the characteristics of corresponding RL algorithms even when optimal policy is not reached.

c. **Experimental Evidence.** UCRL2 is known to be provably regret-optimal and is designed to continuously explore. SAC does not have such rigorous theoretical guarantees but is known to be practically efficient. In Table 1, by UC, we observe that SAC is significantly suboptimal than UCRL2. But SAC has lower ESL than UCRL2 as its exploration is smoother. Additionally, OMR for SAC is higher than that of UCRL2. They together indicate that SAC takes smoother but larger number of policy transitions aligned to optimal direction for exploration, while UCRL2 exhibits bigger policy changes and in diverse manner trying to cover the environment faster.

**2. Complementarity of ESL and OMR with respect to Regret:**

UCRL2 and PSRL have the same order of regret bound (Osband et al., 2013). But PSRL leads to smoother policy transitions that are much more orientated towards optimality (as shown in Figure 3), while UCRL2 leads to less smooth policy transitions that do not taper as it approaches optimality. This information is not evident from regret but from corresponding ESLs and OMRs (Table 1).

**3. Insights for Algorithm Design:**

Knowing ESL (or suboptimal ESL) and OMR can assist with developing algorithms that emphasize certain exploratory characteristics. We can develop algorithms with grades of coverage or directness, while also being able to visualize this. Ultimately, depending on the environment, we can choose which characteristics of exploratory process are well suited. In contrast, looking only at the final outcomes of RL algorithms like regret and number of updates does not include these nuances.

# C    Specifications of the RL Algorithms under Study

## C.1    Methods for simulation results (Discrete MDP)

**Model parameter initialisation.** We initialised model parameters for deep learning RL algorithms like DQN and SAC by uniformly sampling weight values between $-3 \cdot 10^{-4}$ and $3 \cdot 10^{-4}$ and the biases at 0. For tabular Q-learning algorithms, we randomly initialized the Q-values between $-1.0$ and $1.0$. For UCRL and PSRL, the policy model was randomly initialized. Note that all Wasserstein distances were computed using a python package POT (Flamary et al., 2021). Additionally, L1 norm was used in our Wasserstein metric cost function as the ground metric for the 2D gridworld environment.

**Results in Figure 3.** The problem setting was deterministic with dense-rewards and 15 maximum number of steps per episode. The total number of episodes was 200. The convergence criterion was satisfied when maximum returns were produced by an algorithm over 5 consecutive updates. The results showcase a single representative run of each algorithm. The confidence parameter $\delta = 0.1$ was utilized for UCRL2. The $\alpha$ parameter for SAC was autotuned using the approach in (Haarnoja et al., 2019) along with hyperparameters described in Table 9. While DQN began with $\epsilon = 1.0$ and the value decayed as $\epsilon[t + 1] = \max\{0.9999 \times \epsilon[t], 0.0001\}$. Table 10 shows hyperparameters for DQN. Note that the ADAM (Kingma & Ba, 2017) optimizer was used in all the neural network models.

Table 9: SAC Hyperparameters.

| Parameter | Value |
|---|---|
| learning rate | $5 \cdot 10^{-4}$ |
| discount($\gamma$) | 0.99 |
| replay buffer size | $10^4$ |
| number of hidden layers (all networks) | 1 |
| number of hidden units per layer | 32 |
| number of samples per minibatch | 64 |
| nonlinearity | ReLU |
| entropy target | -4 |
| target smoothing coefficient ($\tau$) | 0.01 |
| target update interval | 1 |
| gradient steps | 1 |
| initial exploration steps before model starts updating | 500 |

Table 10: DQN Hyperparameters.

| Parameter | Value |
|---|---|
| learning rate | $5 \cdot 10^{-2}$ |
| discount($\gamma$) | 0.99 |
| replay buffer size | $10^4$ |
| number of hidden layers (all networks) | 1 |
| number of hidden units per layer | 32 |
| number of samples per minibatch | 64 |
| nonlinearity | ReLU |
| target smoothing coefficient ($\tau$) | 0.001 |
| target update interval | 1 |
| gradient steps | 1 |
| initial exploration steps before $\epsilon$ decays | 500 |

**Results in Tables 1 and 2.** The problem settings had 40 maximum number of steps per episode, and the convergence criterion was satisfied when maximum returns were produced by an algorithm over 5 consecutive

updates. The means and standard deviations for each algorithm were computed over 50 runs. The total number of episodes was 200 for results in Table 1, and 500 in Table 2. For results in Figure 5, the Q-learning with decaying $\epsilon$-greedy where $\epsilon = 0.9$ was employed in the gridworld tasks described in Appendix B.1. A convergence criterion of 50 consecutive model updates with maximum returns was utilized. We aggregated the result over 40 training trials and the maximum number of steps per episode was 60.

### C.2 Methods for simulation results (Continuous MDP)

**Model parameter initialisation.** We initialised model parameters for the deep learning SAC algorithm by uniformly sampling weight values between $-3 \cdot 10^{-4}$ and $3 \cdot 10^{-4}$ and the biases at 0. For the DDPG algorithm, the output layer weight values were initialised using Xavier Initialization (Glorot & Bengio, 2010), while the rest were uniformly sampled between $-3 \cdot 10^{-3}$ and $3 \cdot 10^{-3}$. This was done on both the actor and critic networks. The ADAM (Kingma & Ba, 2017) optimizer was used in all the neural network models. In both algorithms, 1) a discount factor $\gamma = 0.99$ was used, 2) 500 initial steps were taken before updating model weights, and 3) replay buffer size was $10^6$. Tables 11 and 12 display hyperparameters for DDPG and SAC, respectively.

**Results in Figure 4.** The problem setting was Mountain Car continuous (Moore, 1990) with 999 maximum number of steps per episode (Brockman et al., 2016). The total number of training episodes was 100. The convergence criterion was satisfied when maximum returns were produced by an algorithm over 10 consecutive updates. The results showcase a single representative run of each algorithm. For results in **Table 3**, the mean and standard deviations for each algorithm were computed over 5 runs. While RL training was conducted in a continuous state-action space, we discretized it for Wasserstein distance calculations between occupancy measures, using 4 bins for actions and 10 bins for states. Note that all Wasserstein distances were computed using a python package POT (Flamary et al., 2021). Additionally, L2 norm was used in our Wasserstein metric cost function as the ground metric for the Mountain Car environment.

Table 11: DDPG Hyperparameters.

| Parameter | Value |
|---|---|
| number of samples per minibatch | 128 |
| nonlinearity | ReLU |
| target smoothing coefficients ($\tau$) | 0.001 |
| target update interval | 1 |
| gradient steps | 1 |
| number of hidden layers (all networks) | 2 |
| number of hidden units per layer | 64 |
| Actor learning rate | $5 \cdot 10^{-4}$ |
| Critic learning rate | $5 \cdot 10^{-3}$ |

Table 12: SAC Hyperparameters.

| Parameter | Value |
|---|---|
| learning rate | $3 \cdot 10^{-3}$ |
| number of hidden layers (all networks) | 2 |
| number of hidden units per layer | 64 |
| number of samples per minibatch | 128 |
| nonlinearity | ReLU |
| target smoothing coefficient ($\tau$) | 0.001 |
| target update interval | 1 |
| gradient steps | 1 |

# D    Supplementary Results

In this section we present enlarged versions of results in Figure 3 (see Section D.1) and additional plots that support the results in the main paper (see Section D.3).

## D.1    Enlarged Visualisation of the Occupancy Measure Trajectories

Figures 9 - 11 are enlarged versions of enlarged versions of Figure 3. For each algorithm, there is a visualisation of the policy trajectory and visualisation of the state visitation below it.

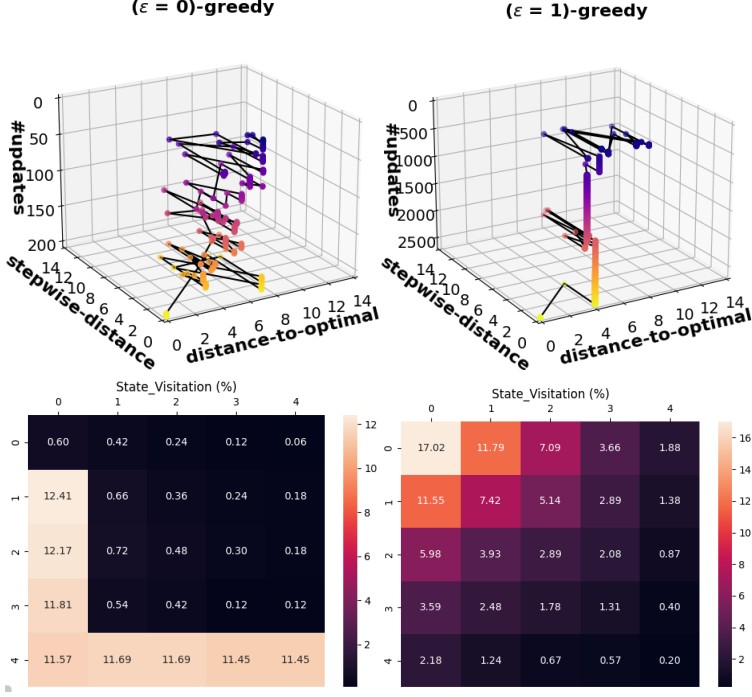

Figure 9: Top row: Scatter plots of *distance-to-optimal* and *stepwise-distance* over updates for $\epsilon(=0)$-greedy and $\epsilon(=1)$-greedy Q-learning. Bottom row: State visitations.

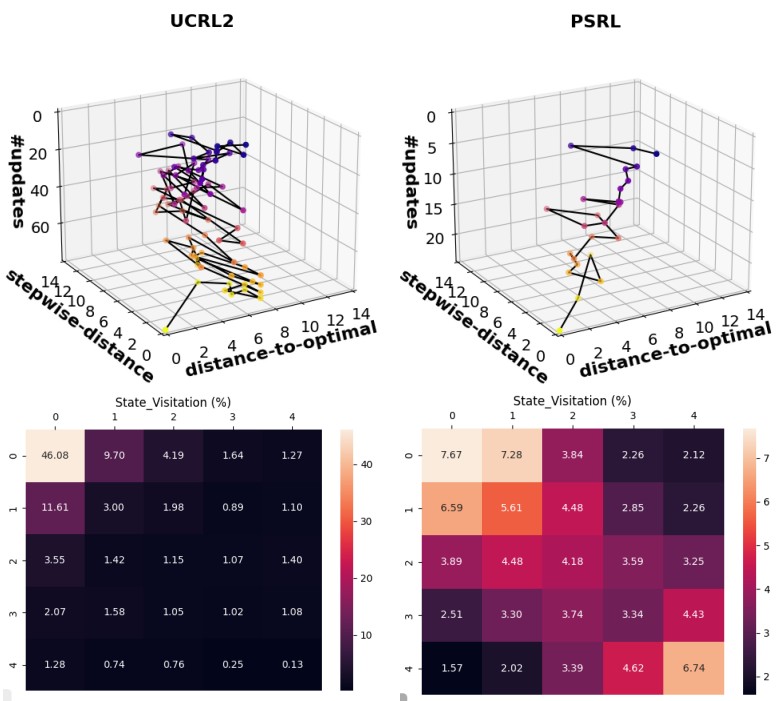

Figure 10: Top row: Scatter plots of *distance-to-optimal* and *stepwise-distance* over updates for UCRL2 and PSRL. Bottom row: State visitations.

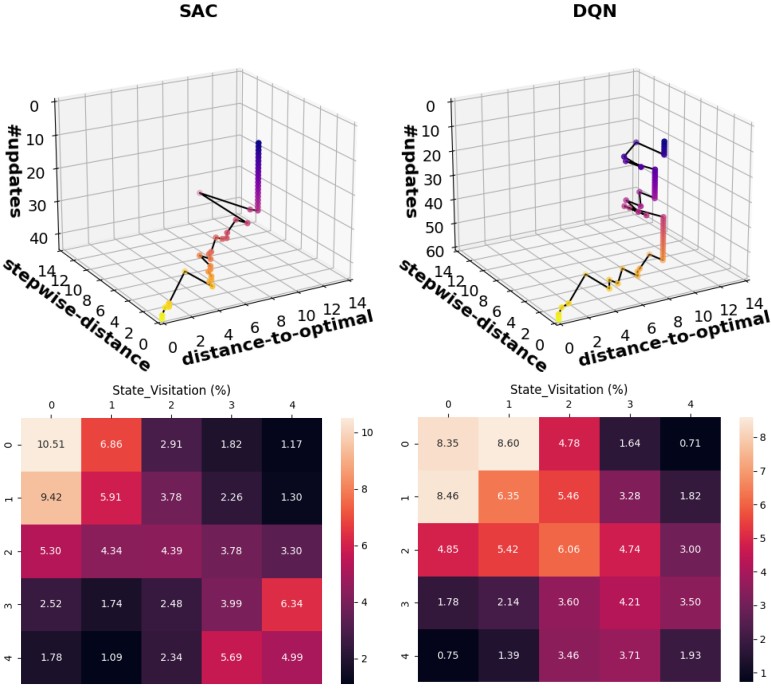

Figure 11: Top row: Scatter plots of *distance-to-optimal* and *stepwise-distance* over updates for SAC and DQN. Bottom row: State visitations.

## D.2 Performance Plots

This section contains Return plots of the algorithms. This allows us to assess the learning of algorithms from the performance perspective. Figures 12 and 13 depict performance evolution that corresponds to settings in Figures 3 and 4, respectively. Note that while all algorithms find the optimal policy, UCRL2 and $\epsilon(=1)$-greedy Q-learning fail to remain there if training continues without truncation. As a result, their performance does not improve over time compared to those that stabilize at the optimal policy.

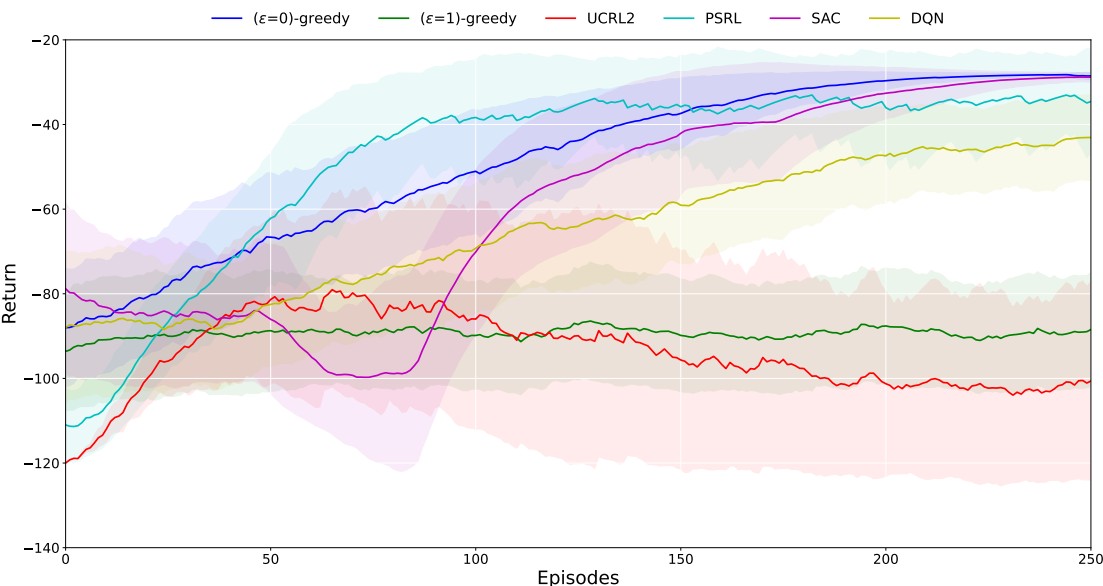

Figure 12: Return plots of algorithms: $\epsilon(=0)$-greedy and $\epsilon(=1)$-greedy Q-learning, UCRL2, PSRL, SAC, and DQN averaged over 5 runs in the deterministic 5×5 Gridworld with dense rewards.

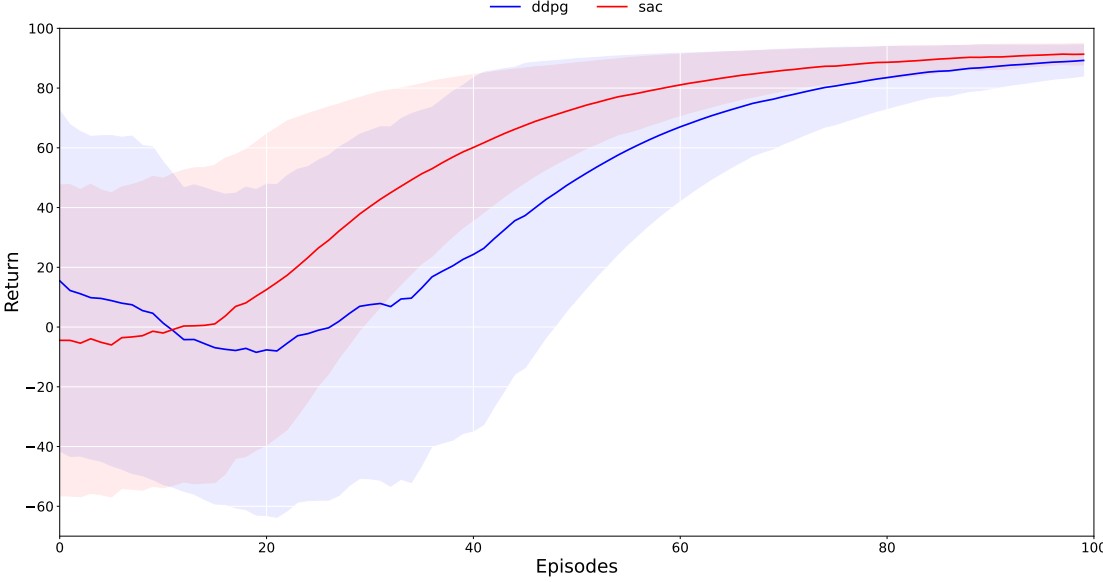

Figure 13: Return plots of algorithms: DDPG and SAC averaged over 5 runs in the continuous Mountain Car problem.

### D.3 Evolution of *stepwise-distance*, *distance-to-optimal*, and OMR($k$)

In this section we present 2 dimensional versions of the policy trajectories in Figures 3 and 4, along with corresponding OMR evolution plots. These are *stepwise-distance* vs. updates, *distance-to-optimal* vs. updates, and OMR($k$) plots for the algorithms. Figure 14 presents plots for the continuous environment Mountain Car, while Figure 15) presents plots for the discrete environment 2D Gridworld.

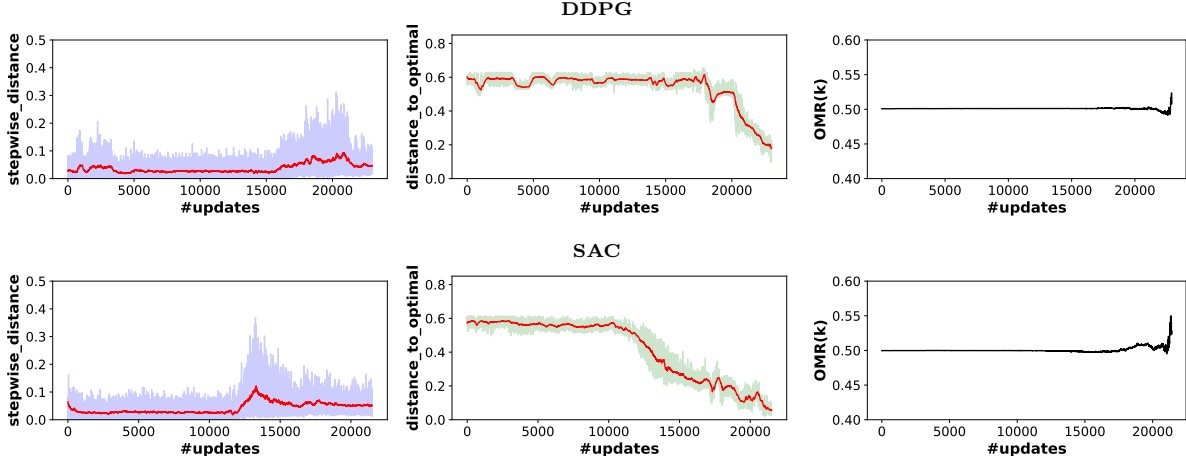

Figure 14: Plots in the first column are *stepwise-distance* vs. number of updates, second column *distance-to-optimal* vs. number of updates, and third OMR($k$) vs. number of updates. Top row plots belong to DDPG algorithm, while bottom row plots belong to SAC.

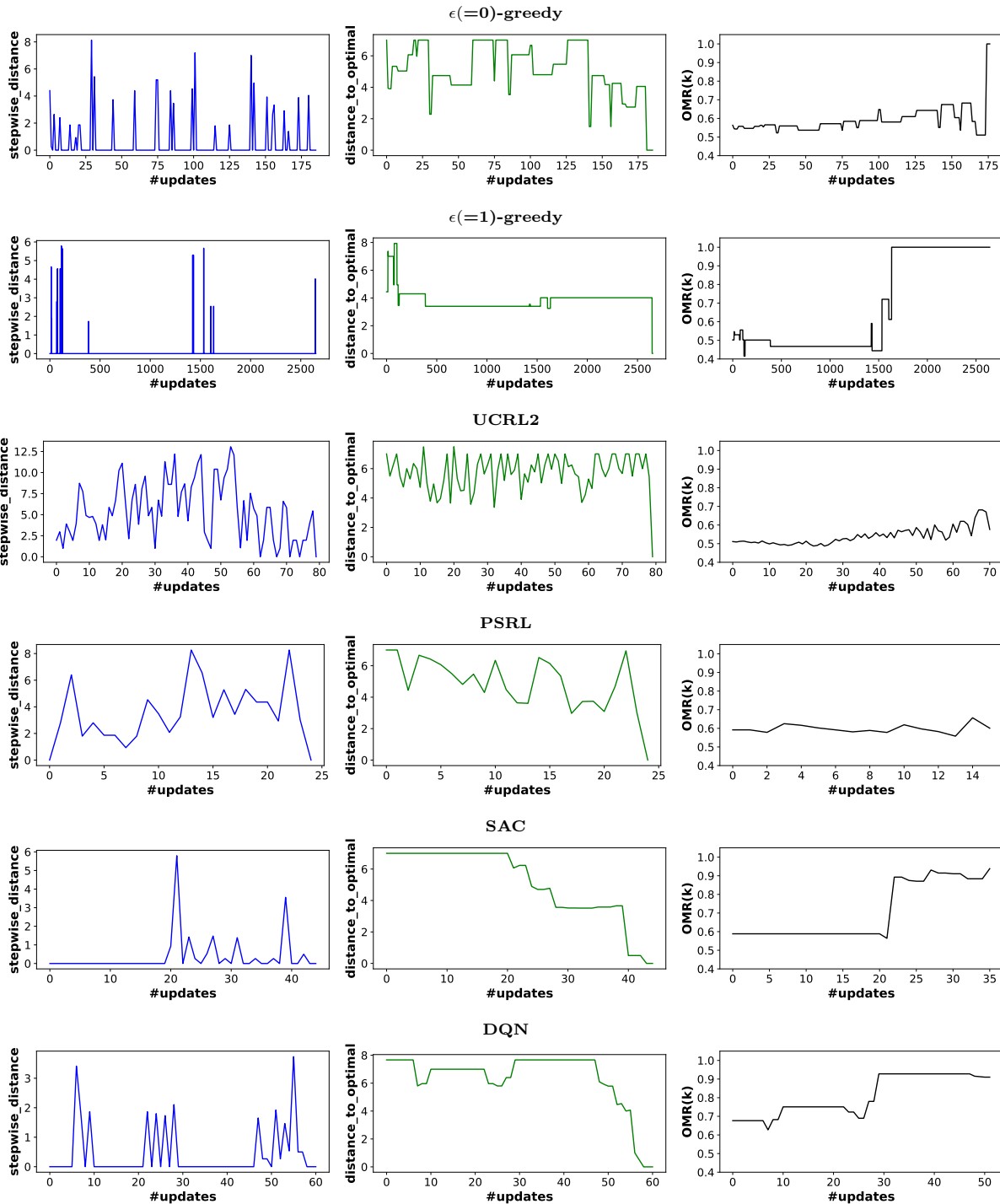

Figure 15: Plots in the first column are *stepwise-distance* vs. number of updates, second column *distance-to-optimal* vs. number of updates, and third OMR($k$) vs. number of updates. The plots in the row belong to algorithms in the following order from top to bottom: $\epsilon(=0)$-greedy, $\epsilon(=1)$-greedy, UCRL2, PSRL, SAC, and DQN.

