# OpenReview forum: "Studying Exploration in RL: An Optimal Transport Analysis of Occupancy Measure Trajectories"
_TMLR — Accepted by TMLR_

### Review · Reviewer_5wrR · 2025-02-25

**Summary Of Contributions:**

1. The paper introduces a geometric framework to analyze exploration in reinforcement learning (RL) by modeling the learning process as trajectories in the space of state-action occupancy measures.
2. It proposes two novel metrics: Effort of Sequential Learning (ESL), which quantifies exploration efficiency as the ratio of an algorithm’s trajectory length to the shortest path (geodesic) between initial and optimal policies, and Optimal Movement Ratio (OMR), which measures the fraction of policy updates that reduce regret-like "distance-to-optimal."
3. Theoretical guarantees are provided for estimating ESL and OMR with finite samples and suboptimal final policies.
4. Empirical validation on discrete and continuous environments demonstrates the metrics’ ability to diagnose exploration efficiency and task difficulty.

**Audience:**

Yes

**Claims And Evidence:**

Yes

**Requested Changes:**

Can the authors demonstrate the reason for using W1 distance? Do the authors also consider W2 distance or KL divergence? How does the choice of distance metrics affect ESL/OMR?

**Typos**
  - Abstract: "finite number of samples" $\rightarrow$ "a finite number of samples."
  - Introduction: "Comparing the exploratory processes of these eclectic algorithms across the multi-directional space of RL algorithm design, emerges as a natural question" includes an unnecessary comma after "design."
  - Keep consistent when using "behaviour" and "behavior"

**Strengths And Weaknesses:**

**Strengths**
1. This work provides a novel geometric perspective on RL exploration with Wasserstein distances, which enables a principled comparison of policy trajectories.
2. This work has a strong theoretical foundation, which links policy dynamics to regret bounds and provides approximation guarantees.
3. The experiments clearly demonstrate ESL/OMR can differentiate exploration behaviors (e.g., direct vs. meandering paths) across algorithms like Q-learning, UCRL2, and SAC.
4. The work is well-written with clear contributions and demonstration.

**Weaknesses**
1. The proposed metrics rely on knowing the optimal policy ($\pi^*$) to compute the geodesic. Approximate $\pi^*$ with suboptimal $\pi_N$ sounds valid but is not validated in complex settings, which may limit applicability to environments where $\pi^*$ is tractable.
2. I am concerned about the computational scalability. Calculating Wasserstein distances between occupancy measures incurs high computational costs, especially for high-dimensional/continuous spaces. Can the authors provide some insights on this?

---

> ### Author Response · Authors · 2025-03-16
>
> We appreciate the reviewer’s thoughtful feedback. Below, we address the identified weaknesses and requested changes in the order they were raised.
>
> **1) Metrics relying on knowing the optimal policy.**
> We appreciate the reviewer's concerns. Indeed, to compute ESL and OMR exactly, we need to know the optimal policy $\pi^{*}$. In the text below Proposition (5), we have clearly indicated the applicability and limitations, as excerpted below.
> >Note that Equation (10) shows that when $\pi_{N}$ is close to $\pi^*$, then $\eta_{sub}$ is a good approximation of $\eta$, and thus a good quantifier to determine the efficiency of the algorithm's exploratory process. However,  $W_{1}(v_{\pi_{N}},v_{\pi^*})$ is dependent on the RL algorithm and hence a bound cannot be provided here. Still, $\eta_{sub}$ might be useful when $\pi_{N}$ is closer to $\pi^{*}$ than $\pi_0$. A fallible proxy for this could be when the performance of $\pi_N$ is better than of $\pi_0$. We show the usefulness of $\eta_{sub}$ in our experimental results in Section 5.3 and Appendix B.5 for simple environments. It remains to be seen how useful $\eta_{sub}$ is in complex environments.
>
> **2) Computational scalability.**
> We appreciate the reviewer's concerns regarding the computational scalability of solving large-scale Optimal Transport (OT) problems inherent in large-scale or high-dimensional environments. We have now added in the Discussion (Section 7):
> >While efficiency is an important aspect, the primary focus of this work is to introduce a framework for analysing exploration in RL using occupancy measures. For large-scale environments, several established methods, such as greedy computation (Carlier et al., 2010), hierarchical methods (Lee et al., 2019), and inexact proximal point methods (Xie et al., 2020) can be used to alleviate computational costs. For example, Gao & Chaudhari (2021) leverages a block-diagonal approximation method to deal with high-dimensional probability distributions similar to ours, and anchor space OT (Huang et al., 2024) addresses multiple OT problems with multiple distributions.
>
> **3) Using W1 distance vs W2 distance and KL-divergence.**
> Thanks for your comment. We have now included these reasons and comparison with $\mathcal{W}_2$ and KL-divergence in our Discussion in Section 7 as below:
> >The $W_1$ satisfies the Kantorovich-Rubinstein duality, making Equation (3) applicable and providing a basis for bounding regret in Proposition 2. In contrast, $W_{p>1}$ does not have such duality. Since $W_1(P,Q) \leq W_{p>1}(P,Q)$ (Villani, 2009), $W_1$ yields tighter regret bounds. Moreover, $W_1$ is less sensitive to outliers and sampling discrepancies than $W_{p>1}$ (Raghvendra et al., 2024), making it well-suited for our setting. Nevertheless, our approach extends to $W_{p>1}$. Compared to KL-divergence, $W_1$ is a metric that satisfies symmetry and triangle inequality, which have been instrumental in our proofs and guarantees, while KL-divergence is not a metric. Additionally, $W_{p\geq1}$ leverages the geometry of the underlying support space (Peyré, 2019). This allows it to capture distances between distributions with disjoint support (Peyré, 2019), which KL-divergence cannot capture. Hence, we use $W_1$ in our approach.
>
> **4) Impact of the choice of distance metrics on ESL/OMR.**
> We thank the reviewer for the question. Indeed, the choice of metric should reflect the effort of moving
> in the underlying state space. Thus, we used L1 distance in Gridworld and L2 distance in Mountain Car
> environment. While describing our environments, in Section 5, we have now added:
> >Note that we used $L1$ distance and $L2$ distance as metrics ($d_{X}$) for the state spaces of the 2D-Gridworld and Mountain Car, respectively, which underpin the $W_1$.
>
> In the Discussion (Section 7), we have added:
> >The choice of the distance metric impacts the geometry of the support space (Lee, 2009),
> consequently the Wasserstein distances, and thus ESL and OMR. In our case, the support space (i.e. state-action space) is reduced to the state space, as the action space maps back to the state space via OTDD. Thus, the choice of distance metric should reflect the effort of moving in the state space. For example, in the Gridworld, we used L1 distance because only vertical and horizontal displacements are allowed, and L2 distance in the Mountain Car, as applicable to real-world spaces.

---

> > ### Comment · Reviewer_5wrR · 2025-04-02
> > **Official Comment by Reviewer 5wrR**
> >
> > I appreciate the authors' response. My concerns have been addressed, and I would recommend accepting this work.

---

### Review · Reviewer_rnie · 2025-03-03

**Summary Of Contributions:**

This paper introduces a novel framework for analyzing and comparing reinforcement learning algorithms based on the trajectories they take in the space of state-action occupancy measures. The key insight is that any RL algorithm generates a sequence of policies during training, which can be viewed as a path in the manifold of occupancy measures.
The authors define two primary metrics:
- **ESL (Effort of Sequential Learning):** Measures the ratio of the path traversed by an RL algorithm to the direct distance between initial and optimal policies
- **OMR (Optimal Movement Ratio):** Measures the proportion of policy updates that effectively reduce the distance to optimal policy

The paper establishes a mathematical foundation by proving that the space of occupancy measures forms a differentiable manifold for smoothly parameterized policies. They also connect their metrics to regret, showing that regret is related to the sum of distances between the optimal policy and each policy in the learning sequence.

For practical implementation, the authors derive approximation guarantees for estimating ESL and OMR with finite samples, including when the optimal policy is not reached. Through experiments across various environments (discrete and continuous) and algorithms (Q-learning, UCRL2, PSRL, SAC, DQN, DDPG), they demonstrate that their metrics provide insights into exploration processes and task difficulty.

**Audience:**

Yes

**Broader Impact Concerns:**

I don't think this work has ethical issues.

**Claims And Evidence:**

Yes

**Requested Changes:**

In general, I think this paper provides interesting insights into a new method of comparing algorithms and tasks in RL problems. Regarding each weakness I mentioned, I suggest the authors make changes correspondingly.
1. **Computational Complexity:** Give details of how to approximate Wasserstein distance when state-action space is large.
2. **Dependence on Optimal Policy:** Make clear arguments about the applicable scenarios when Proposition 5 gives meaningful approximation factor. Say, how can we guarantee $\mathcal{W}\_1 (v_{\pi_N}, v_{\pi^*}) < c$ for some constant $c$?
3. **Limited Experiments:** Maybe add experiments on harder tasks. This can be related to the first weakness.
4. **Strong Assumptions:** Verify that for the simple environments in Section 5, such assumption holds (or not).
5. **Limited Practical Guidelines:** Add more discussions on how will ESL and OMR related to algorithm design for RL tasks.

**Strengths And Weaknesses:**

### Strengths

1. **Novel Theoretical Framework:** The paper introduces a mathematically rigorous framework for understanding RL algorithms through the lens of occupancy measures and optimal transport, providing a new perspective on exploration. The two metrics, ESL and OMR, proposed in this paper are novel in the literature.
2. **Impacts other than Comparing Algorithms:** By connecting their occupancy measure-based metrics to regret (a standard performance metric), the authors bridge theoretical understanding with practical performance evaluation. The paper also demonstrates that ESL scales proportionally with task difficulty, providing a way to quantify the relative hardness of different environments.
3. **Consideration of Practical Cases:** The authors provide methods to estimate ESL and OMR, and give upper bound on the estimation error. Moreover, when the optimal policy is not reached, the surrogate quantity $\eta_{\textup{sub}}$ for ESL can also be bounded near the true value.

### Weaknesses

1. **Computational Complexity:** Computing the Wasserstein distance between occupancy measures can be computationally intensive, especially for large state-action spaces, making practical application challenging.
2. **Dependence on Optimal Policy:** While the authors provide methods to approximate their metrics when the optimal policy is unknown, the most accurate measurements still require knowledge of the optimal policy, which is rarely available in real-world applications. Meanwhile, though Proposition 5 gives an approximation factor of $\eta_{\textup{sub}}$ with respect to $\eta$, this multiplicative approximation is not very informative, as $\mathcal{W}\_1 (v_{\pi_N}, v_{\pi^*})$ may be large when $\pi_N$ is far away from $\pi^*$. This results in high-level requires the algorithm to reach a neighborhood of the optimal policy to estimate $\eta_{\textup{sub}}$ accurately.
3. **Limited Experiments:** The experiments are primarily conducted on relatively simple environments (GridWorld and Mountain Car). More complex domains with high-dimensional state spaces would better demonstrate the framework's scalability.
4. **Strong Assumptions:** The assumption on existence of inverse transition matrices (Proposition 1) may not hold in practical settings. The authors didn't verify that for the simple environments in Section 5, such assumption holds.
5. **Limited Practical Guidelines:** While the metrics provide insights into exploration processes, the paper doesn't explicitly guide how to design better RL algorithms based on these insights.

---

> ### Author Response · Authors · 2025-03-16
>
> We appreciate the reviewer's thoughtful feedback and address both the weaknesses and requested changes together, as they fall under the same headings.
>
> **1. Computational Complexity & Scalability.**
> We appreciate the reviewer's concerns regarding the computational scalability of solving large-scale Optimal Transport (OT) problems inherent in large-scale or high-dimensional environments. We have now added in the Discussion (Section 7):
> >While efficiency is an important aspect, the primary focus of this work is to introduce a framework for analyzing exploration in RL using occupancy measures. For large-scale environments, several established methods, such as greedy computation (Carlier et al., 2010), hierarchical methods (Lee et al., 2019), and inexact proximal point methods (Xie et al., 2020) can be used to alleviate computational costs. For example, Gao & Chaudhari (2021) leverages a block-diagonal approximation method to deal with high-dimensional probability distributions similar to ours, and anchor space OT (Huang et al., 2024) addresses multiple OT problems with multiple distributions.
>
> **2. Dependence on Optimal Policy.**
> We appreciate the reviewer's concerns. Indeed, to compute ESL and OMR exactly, we need to know the optimal policy $\pi^{*}$. In the text below Proposition (5), we have clearly indicated the applicability and limitations, as excerpted below.
> >Note that Equation (10) shows that when $\pi_{N}$ is close to $\pi^*$, then $\eta_{sub}$ is a good approximation of $\eta$, and thus a good quantifier to determine the efficiency of the algorithm's exploratory process. However,  $W_{1}(v_{\pi_{N}},v_{\pi^*})$ is dependent on the RL algorithm and hence a bound cannot be provided here. Still, $\eta_{sub}$ might be useful when $\pi_{N}$ is closer to $\pi^{*}$ than $\pi_0$. A fallible proxy for this could be when the performance of $\pi_N$ is better than of $\pi_0$. We show the usefulness of $\eta_{sub}$ in our experimental results in Section 5.3 and Appendix B.5 for simple environments. It remains to be seen how useful $\eta_{sub}$ is in complex environments.
>
> **4. Strong Assumptions.**
> We thank the reviewer for highlighting the assumption regarding the existence of inverse transition matrices. Upon carefully reviewing Equations 27 and 28 (in Appendix A.3), we found this assumption to be unnecessary. We have re-derived Equation 28 to eliminate its dependency on the inverse transition matrix (modified in Appendix A.3). As a result, the revised Proposition 1 is now stated as:
> >If the policy $\pi$ has a smooth parameterization $\theta$, then the space of occupancy measures $\mathcal{M}$ is a differentiable manifold.
>
> Similarly, modification were made in Equation 37 (in Appendix A.5) and the revised Proposition 3 is updated to:
> >If the policy $\pi$ has a smooth parametrization $\theta$, then the space of finite-horizon occupancy measures $\mathcal{M}^H$ is a differentiable manifold.
>
> **5. Limited Practical Guidelines.**
> We thank the reviewer for this suggestion. We have now added in the Discussion (Section 7):
>
> >Depending on the environment, we can select an algorithm with promising characteristics and spend more time optimising it to improve performance. For example, rather than tuning the hyper-parameters of multiple competing algorithms to find the best one, it may be more effective to first identify an algorithm best suited to an environment based on ESL and OMR, and then fine-tune it. The chosen algorithm might remain suitable across similar environments as well.
> Furthermore, we could incorporate an online adaptation of the exploratory process of the RL algorithm itself, based on recent estimates of the ESL and OMR. For example, if the current policy gives a better return than the initial policy, then we could adapt the exploratory parameters (at a slow rate) to optimise a running estimate of  $\eta_{sub}$ and a suitable approximation of OMR, thus enabling better exploration. However, the feasibility and convergence of such a scheme remains open.

---

> > ### Comment · Reviewer_rnie · 2025-03-23
> > **Response to rebuttal**
> >
> > The rebuttal addresses most of my concerns, and I'm looking forward to see more experiment results. I understand that adding more experiments may demand more time and compute, so I'm satisfied with current results.

---

### Review · Reviewer_CbTT · 2025-03-04

**Summary Of Contributions:**

This paper proposes a framework for analyzing the learning process in reinforcement learning. By representing algorithms as trajectories of occupancy measures, it defines two metrics—Effort of Sequential Learning (ESL) and Optimal Movement Ratio (OMR)—to quantify exploration efficiency. The authors establish theoretical guarantees and provide empirical evidence that these metrics capture meaningful insights into exploration behavior. The most interesting point is that this paper provides a new perspective on analyzing the characteristics of algorithms, which is promising and worthy of further research.

**Audience:**

Yes

**Broader Impact Concerns:**

This paper does not have any concerns on the ethical implications.

**Claims And Evidence:**

Yes

**Requested Changes:**

This paper is generally a good and interesting paper that provide a framework to visualize and analyze the policy evolution during RL training, thus giving a new perspective on analyzing the characteristics of algorithms. **I recommend accepting this paper**.

However, I still have some suggestions for this paper and some problems need to be clarified from the authors:

1. As is stated in weaknesses (3), it is interesting to discuss whether the summation of distance-to-optimal can become a similar evaluation way to regret of an algorithm. I suggest adding some further discussion on this point.

2. In page 5, how to understand the statement "we can show that $v_\pi^H$ satisfies the linear programming description of value function maximization along with the Bellman flow constraints"?

3. Is it necessary to pre-know optimal policy for the calculation of ESL and OMR? In experiment part, is the optimal policy pre-known? (Otherwise we cannot calculate distance-to-optimal while training). If so, I think the reliance on knowledge of an optimal policy will complicate the framework’s practical usage. **I hope the author can clearly explain this question.**

4. Does this framework only work in MDP with $L_R$-Lipschitz reward setting? Can it be applied in more general settings?

5. The experiments to analyze the characteristics of algorithms are interesting. I suggest implementing on more popular algorithms to make the paper more convincing. Moreover, I think the 3D scatter plots in Figure 3 is not very intuitional though it conveys some phenomenon. I suggest the author also providing projection plots (i.e. y-axis is updates and x-axis is stepwise-distance or distance-to-optimal), which may be more direct for readers to understand the characteristics of algorithms.

6. Drawing Q-functions plot is a common way to record the performance of algorithms. I suggest the author also provide Q-function plot to compare with existing plots. I would like to know the unique advantages of your provided methods to analyze algorithms' characteristics.

7. The paper states that "Ultimately, depending on the environment, we can select an algorithm with promising characteristics and spend more time optimizing it to improve performance." Can you explain more about this idea? Do you mean that we can use this framework to analyze characteristics of algorithms first then choose the most promising algorithm that can best adapt the environment? Will the characteristics will change if the environment changes?

8. In the definition of $\mathcal{M}$, that is, $\mathcal{M}=\lbrace v_{\pi_\theta}(s, a) \mid \pi_\theta \in \Gamma_\theta, \theta \in \mathbb{R}^{N_\theta} \rbrace$ in page 3. I don't think there should be a $\theta$ in policy set $\Gamma$. Is this a typo?

**Strengths And Weaknesses:**

Strengths:
1. This paper give an interesting framework to visualize and analyze the policy evolution during RL training.

2. The paper rigorously formalizes occupancy measures as points on a differentiable manifold and employs the Wasserstein distance to measure trajectories, providing a strong theoretical underpinning.

3. ESL and OMR are introduced to help analyze the characteristics of algorithms, which is a different evaluation perspective from regret. This direction is promising and worthy of further research.

4. The experiment part is precise and well-explained, making the advantages and functions of this proposed framework clear.


Weaknesses:

1. **Reliance on knowledge of an optimal policy** may hinder the direct computation of distance-to-optimal, which can complicate the framework’s practical usage.

2. Estimating Wasserstein distance and occupancy measures can be expensive for large-scale or high-dimensional environments, potentially limiting scalability.

3. It's good that Proposition 2 gives a connection between regret and occupancy measure, but further analysis is lacking. For example, it is interesting to discuss whether the summation of distance-to-optimal can become a similar evaluation way to regret of an algorithm.

---

> ### Author Response · Authors · 2025-03-16
>
> We appreciate the reviewer’s thoughtful feedback and their positive evaluation of the paper. Below we address the requested changes in the order they were raised.
>
> **1) Connection between regret and occupancy measure.** Regret is the difference between the maximum reward that could have been achieved and the actual collected rewards during training. However, the *sum of distance-to-optimal* captures the difference between the optimal behaviour and the actual behaviour of the agent during training. Proposition 2 provides a bound between the two quantities. In the discussion, Section 7, we added:
>
> >Comparing regret across policies measures the similarity of returns (with respect to optimal), disregarding behavioural differences, like variations in actions at the same states. This makes regret advantageous in settings without critical safety and physical constraints, e.g. games, due to its computational efficiency. In contrast, the *sum-of-distance-to-optimal* focuses on behavioural differences between policies, is reward-agnostic, and is well-suited for environments where safety and physical constraints are critical, e.g. robotics. While minimizing regret prioritizes matching the performance of the optimal policy, minimizing the *sum-of-distance-to-optimal* focuses on mimicking its behaviour. Thus, the *sum-of-distance-to-optimal* can be used similarly to regret, especially where the behaviour of achieving good performance is essential.
>
> **2)**
> In Section 3.3, we have replaced the aforementioned statement by:
> >Following Syed et al. (2008), work by Kalagarla et al. (2021) shows that $v^H_{\pi}$ can be used in the linear programming formulation for solving MDPs and satisfies the Bellman Flow Constraint (in Equation (19) from Appendix A.3).
>
> **3) Metrics relying on knowing the optimal policy.**
> We appreciate the reviewer's concerns. It is necessary to pre-know the optimal policy when computing ESL and OMR, similar to regret. In these simple experiments, we know the optimal policy/policies and we confirmed that the final policy is optimal, and computed ESL and OMR. In the text below Proposition (5), we have clearly indicated the applicability and limitations, as excerpted below.
> >Note that Equation (10) shows that when $\pi_{N}$ is close to $\pi^*$, then $\eta_{sub}$ is a good approximation of $\eta$, and thus a good quantifier to determine the efficiency of the algorithm's exploratory process. However,  $W_{1}(v_{\pi_{N}},v_{\pi^*})$ is dependent on the RL algorithm and hence a bound cannot be provided here. Still, $\eta_{sub}$ might be useful when $\pi_{N}$ is closer to $\pi^{*}$ than $\pi_0$. A fallible proxy for this could be when the performance of $\pi_N$ is better than of $\pi_0$. We show the usefulness of $\eta_{sub}$ in our experimental results in Section 5.3 and Appendix B.5 for simple environments. It remains to be seen how useful $\eta_{sub}$ is in complex environments.
>
> **4)**
> Our framework is applicable to any MDP reward settings, however the relation between regret and the *sum of distance-to-optimal* in the occupancy measure space (i.e. Proposition 2) is guaranteed for $L_R$-Lipschitz reward settings.
>
> **5)**
> We appreciate the reviewer's suggestions. We show the projections of the plots in 2D in Appendix D.3 as Figures 13 and 14 and have indicated this in the caption of Figure 3. Regarding popular algorithms, we have implemented several widely recognized algorithms including SAC, DDPG, DQN, and Q-learning, as well as UCRL2 and PSRL from the RL theoretical perspective. We would be grateful if the reviewer could suggest any additional algorithms they consider particularly relevant.
>
> **6)**
> Thank you for the suggestion. Could the reviewer clarify what they are referring to by Q-function plots? If they mean plots of Return vs training episodes, we have now included these Return plots for each algorithm during training in Appendix D.2 (Figures 11 and 12). If not, we would appreciate further details.

---

> > ### Author Response · Authors · 2025-03-16
> >
> > **7)**
> > We thank the reviewer for this question. We have now added in the Discussion (Section 7):
> >
> > >Depending on the environment, we can select an algorithm with promising characteristics and spend more time optimising it to improve performance. For example, rather than tuning the hyper-parameters of multiple competing algorithms to find the best one, it may be more effective to first identify an algorithm best suited to an environment based on ESL and OMR, and then fine-tune it. The chosen algorithm might remain suitable across similar environments as well.
> > Furthermore, we could incorporate an online adaptation of the exploratory process of the RL algorithm itself, based on recent estimates of the ESL and OMR. For example, if the current policy gives a better return than the initial policy, then we could adapt the exploratory parameters (at a slow rate) to optimise a running estimate of  $\eta_{sub}$ and a suitable approximation of OMR, thus enabling better exploration. However, the feasibility and convergence of such a scheme remains open.
> >
> > **8)**
> > We thank the reviewer for the observation. We have modified the statement (in Section 3) to:
> >
> > >We assume these policies belong to a set of stationary Markov policies parameterised by $\theta \in \Theta$. For policies in this set $\pi_{\theta} \in \Gamma_{\Theta}$, we define the space of occupancy measures corresponding to $\Gamma_{\Theta}$ as $\mathcal{M}$ = {$ v_{\pi_{\theta}}(s,a) \mid \pi_{\theta} \in \mathbf{\Gamma}_{\Theta}, \theta \in \Theta $}

---

> > > ### Comment · Reviewer_CbTT · 2025-03-30
> > >
> > > Thank you for the detailed response, my questions have been addressed.

---

### Decision · Action_Editor_Un4q · 2025-04-14

**Recommendation:** Accept with minor revision

**Comment:**

This paper proposes a framework for analyzing the learning process in reinforcement learning by representing algorithms as trajectories over occupancy measures. It introduces two novel metrics—Effort of Sequential Learning (ESL) and Optimal Movement Ratio (OMR)—to quantify exploration efficiency. The authors provide theoretical guarantees and empirical evidence that these metrics offer meaningful insights into exploration dynamics.

A particularly interesting aspect of the work is its new perspective on understanding the behavior of learning algorithms through occupancy measure trajectories and the use of Wasserstein distance. This approach opens up a promising direction for further research in characterizing and comparing RL algorithms beyond standard performance metrics.

The authors have addressed most of the reviewers’ concerns in their response. One remaining point is that both ESL and OMR require knowledge of the optimal policy. While the authors draw an analogy to regret and discuss optimal policy approximation in Section 4.2, this comparison is not entirely precise: regret can be practically visualized using cumulative rewards, whereas the proposed metrics depend more directly on the unknown optimal policy and lack a straightforward empirical proxy. It would be beneficial if the authors could explicitly discuss this limitation where the metrics are defined, in Section 3.

Despite this minor concern, the paper is well-structured and provides a thoughtful and theoretically grounded contribution to the understanding of learning dynamics in RL. I recommend acceptance with minor revisions to address the above clarification.

**Audience:**

This paper has a potential impact on many researchers in the empirical evaluation of RL algorithms.

**Claims And Evidence:**

Yes, this paper provides insights into the exploration processes of RL algorithms and hardness of different tasks via proposing new metrics based on the learning process of an RL algorithm.

---

> ### Author Response · Authors · 2025-05-06
>
> We appreciate the Action Editor’s decision to accept our paper along with a thoughtful feedback. To address the requested change in the paper, we have now added the statement below in Section 3.2 (after definitions of our metrics). If you approve, then we will be happy to submit the camera-ready version straight away. Thank you.
>
> >The definitions of ESL and OMR assume that the policy at the end of learning is optimal. In Section 4.2, we define a version of ESL that is useful for the cases where an optimal policy is not reached. While this is not an empirical proxy, we show in Section 5.3 and Appendix B.5 that it is useful when the final policy is closer to optimal than the initial one. While regret also depends on an optimal policy, it is related to cumulative rewards, whereas our metrics do not explicitly depend on rewards. Still, we show a bound with regret in Proposition 2, and further discuss the possibility of extending our metrics to be reward-aware in Section 6. We show empirically that our metrics are complementary to regret in Section 5.2, and discuss other connections with regret in Section 7.

---

> > ### Comment · Action_Editor_Un4q · 2025-05-06
> >
> > Thank you for this candid discussion regarding the proposed metrics. The clarity has significantly improved. I approve this change and encourage you to submit the final version.